# SPATIA: MULTIMODAL MODEL FOR PREDICTION AND GENERATION OF SPATIAL CELL PHENOTYPES

## ABSTRACT

Understanding how cellular morphology, gene expression, and spatial organization jointly shape tissue function is a central challenge in biology. Image-based spatial transcriptomics technologies now provide high-resolution measurements of cell images and gene expression profiles, but existing methods typically analyze these modalities in isolation or at limited resolution. We address the problem by introducing SPATIA, a multi-scale generative and predictive model that learns unified, spatially aware representations by fusing morphology, gene expression, and spatial context from single-cell to tissue level. SPATIA also incorporates a spatially conditioned image-to-image generation module that predicts cell morphologies under perturbations, enabling the study of microenvironment-dependent morphological changes such as tumor progression, immune remodeling, and subtype transitions. We assembled a multi-scale dataset consisting of 17 million cell-gene pairs, 1 million niche-gene pairs, and $10,000$ tissue-gene pairs across diverse tissues and disease states. We benchmark SPATIA against 16 existing models across 12 individual tasks, which span several categories including cell phenotype generation, cell annotation, cell clustering, gene imputation, and cross-modal prediction. SPATIA achieves improved performance over baselines and generates realistic cell morphologies that reflect transcriptomic perturbations.

## 1 INTRODUCTION

Understanding the interplay between cellular morphology, gene expression, and spatial organization is essential for modeling tissue function and cell states in health and disease (Szałata et al., 2024; Stirling et al., 2021). Image-based spatial transcriptomic (ST) technologies enable high-resolution profiling of gene expression in intact tissue, along with matched cellular morphology derived from microscopy images (Ståhl et al., 2016; Chen et al., 2015; Janesick et al., 2023; Li et al., 2024c). However, existing approaches often analyze morphology and gene expression separately, which limits their ability to learn representations of cellular phenotypes within spatial context. The central challenge is to learn unified representations that (i) capture the joint structure between image and gene modalities (Chelebian et al., 2025; Min et al., 2024), (ii) preserve spatial dependencies at the single-cell level (Birk et al., 2025; Wen et al., 2023), and (iii) generalize across scales from local niches to whole-slide tissue context (Schaar et al., 2024).

Naive fusion strategies, such as concatenating gene expression vectors with image features or training separate unimodal models, have limited ability to capture nonlinear and context-dependent relationships between modalities (Li et al., 2024b). These limitations are amplified in spatial omics, where cellular identity and state are determined not only by intrinsic features but also by neighboring cells and broader tissue architecture. Existing models fall short in integrating spatial, molecular, and morphological information at single-cell resolution. Single-cell foundation models focus on transcriptomics and ignore morphology entirely (Cui et al., 2023; Kalfon et al., 2025) or focus on spot-level spatial correlations (Tian et al., 2024; Wang et al., 2025a; Wen et al., 2023; Schaar et al., 2024; Li et al., 2025). Pathology models (Chen et al., 2022; 2024b) excel at whole-slide image analysis but disregard molecular information. Vision-language models (Huang et al., 2023; Lu et al., 2024; Ding et al., 2024) rely on textual supervision and cannot model image-gene relationships or spatial dependencies. Recent multimodal ST models (Lin et al., 2024; Chen et al., 2024a) aim to align histology with transcriptomics, but operate only at patch or spot resolution and lack single-cell granularity. As highlighted in recent evaluations of multimodal LLMs for vision-language reason-

ing (Huang et al., 2023; Lu et al., 2023), these models struggle with grounding in spatial structure, compositional reasoning, and fine-grained biological semantics.

These limitations span three dimensions. First, they fail to capture the full range of morphological variation and gene expression patterns at single-cell resolution, which is essential for understanding cell identity, state, and function. Second, they do not model spatial interactions across scales. Biological processes are governed not only by individual cell properties but also by local neighborhoods (niches) and tissue-level organization. Capturing these dependencies requires models that integrate cell-intrinsic features with context-aware representations across multiple spatial levels. Third, current methods cannot accurately predict how cell morphology changes under perturbations in a spatially dependent manner. Unlike generic image synthesis, spatial morphology prediction is challenging: cellular responses to perturbations depend strongly on microenvironmental context, including exposure to signaling

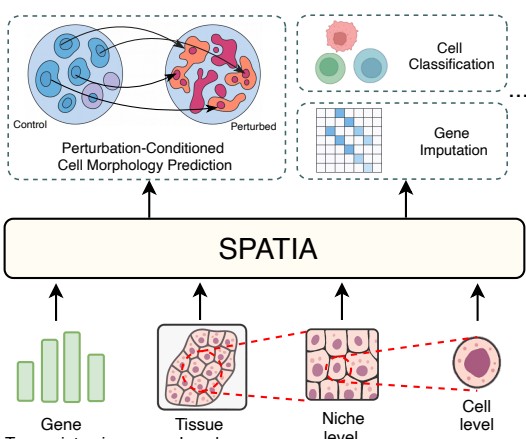

Figure 1: SPATIA is a multi-scale spatial model for predictive and generative tasks.

molecules, immune surveillance, and cell–cell interactions. Modeling these effects requires generative approaches that can respect both the intrinsic state of the cell and its extrinsic spatial niche.

**Present work.** We introduce SPATIA, a multi-level model for generative and predictive modeling of spatial cell phenotypes (Fig. 1). SPATIA integrates cell morphology, gene expression, and spatial coordinates within a unified model. The model consists of three components. At the *cellular level*, SPATIA fuses image-derived morphological tokens and transcriptomic embeddings using cross-attention to produce a single-cell representation that captures both visual and molecular features. At the *niche level*, SPATIA groups neighboring cells into spatial patches (e.g., 256×256 pixels) and applies a transformer to model local cell-cell interactions. At the *tissue level*, a global transformer aggregates niche representations to capture long-range dependencies across the full slide. Each instance links morphology and gene expression at matched spatial scales, enabling fine-grained multimodal representation learning.

SPATIA introduces a spatially conditioned image-to-image generation module designed to predict morphological outcomes of perturbations within tissue context. We form weak data pairs of control and perturbed cells of the same type within spatially adjacent or niche-consistent regions, using optimal transport alignment to balance distributions across states. For generation, we employ flow matching conditioned on the unified cell embedding of SPATIA and perturbation embedding derived from pathology state labels or differential gene expression signatures. A contrastive flow objective further improves state-specific guidance, preserving cell identity while introducing perturbation-specific changes. This design allows SPATIA to simulate microenvironment-dependent morphological changes such as DCIS-to-invasive progression and immune-cold to immune-hot remodeling.

SPATIA is trained on MIST (Multi-scale dataset for Image-based Spatial Transcriptomics), a newly assembled multi-level dataset of image-based spatial transcriptomics. MIST (MIST-C-17M, MIST-N-1M, MIST-T-10K) contains 17 million cell-gene pairs, 1 million niche-gene pairs, and 10,000 tissue-gene pairs, spanning 49 donors, 17 tissue types, and 12 disease contexts. Our code is available at https://anonymous.4open.science/r/upload-2488/README.md

Our main contributions include: ① **Joint modeling of cell morphology and gene expression** We align morphological tokens with transcriptomic embeddings at the single-cell level. This yields unified embeddings that preserve modality-specific detail while capturing their contextual relationships. ② **Multi-level spatial context modeling**. We develop a hierarchical transformer architecture that encodes spatial dependencies at the cell, niche, and tissue levels. This design enables modeling of both local cell-cell interactions and long-range tissue organization within a single unified model. ③ **Predict cell morphologies under perturbation.** By conditioning on unified cell embeddings and pseudo-perturbation (state labels and differential expression signatures), SPATIA captures

morphological changes that are specific to microenvironmental context, enabling a niche-informed framework for conditional morphology synthesis in spatial transcriptomics. ④ **New assembled dataset**. We construct and release MIST, a new dataset for spatial transcriptomics containing 17 million cell-gene, 1 million niche-gene, and $10,000$ tissue-gene pairs across 49 donors, 17 tissue types, and 12 disease contexts, with one-to-one mappings between morphology and transcriptomic profiles, including data from both DAPI and H&E staining. ⑤ **Evaluation across modalities and scales**. We benchmark SPATIA against 16 existing models across 12 individual tasks, which span several categories including cell annotation, cell clustering, gene imputation, and image generation. SPATIA outperforms baselines across scales and modalities.

## 2 RELATED WORK

**Spatial Transcriptomics Models.** Recent models include scGPT-spatial (Wang et al., 2025a), which continually pretrains an scRNA-seq model on multiple platforms of spatial data; CellPLM (Wen et al., 2023), a transformer-based cell language model pretrained on spatially resolved transcriptomic data to encode inter-cell relations; Methods such as SpaGCN (Hu et al., 2021), STAligner (Zhou et al., 2023), and SpaOTsc (Cang & Nie, 2020) integrate spatial transcriptomics with histology, but primarily at spot- or patch-level rather than true single-cell multimodality. SpaGCN and STAligner operate on Visium-like spots to identify spatial domains or align datasets, and SpaOTsc maps scRNA-seq profiles to spatial references without using morphology. Additionally, most methods operate at spot-level resolution (Vicari et al., 2024; Tian et al., 2024; Yang et al., 2025; Wang et al., 2024), lack single-cell granularity, and neglect integration of high-resolution histology or full-slide spatial context.

**Computational Pathology Models.** Vision-only models, such as HIPT (Chen et al., 2022) and UNI (Chen et al., 2024b), utilize hierarchical and self-supervised ViT pretraining on gigapixel WSIs. Vision-language approaches such as CONCH (Lu et al., 2024) and TITAN (Ding et al., 2024) employ contrastive and generative alignment with captions and reports to enable retrieval and report generation. Multimodal image-omic models such as ST-Align (Lin et al., 2024), STimage-1K4M (Chen et al., 2024a), HEST-1k (Jaume et al., 2024) integrate spatial transcriptomics and morphology for gene expression inference and cell mapping. However, existing models are also constrained to spot-level resolution and do not capture single-cell granularity, which is crucial for dissecting cellular heterogeneity and microenvironmental interactions. Vision-only models lack explicit neighborhood or multi-scale tissue context, whereas vision-language models heavily depend on textual annotations, which can vary in quality.

**Generative Models.** Diffusion-based (Ho et al., 2020; Dhariwal & Nichol, 2021) and flow-matching-based generative models (Lipman et al., 2022) are powerful frameworks that transform noise into structured outputs, enabling high-fidelity and conditional synthesis. In the biomedical domain, such models have been increasingly applied to capture the complexity of cellular systems. For example, cellular morphology painting (Navidi et al., 2025), gene expression prediction (Huang et al., 2025b;a; Zhu et al., 2025), Simulating Cellular Morphology Changes (Zhang et al., 2025; Wang et al., 2025b; Palma et al., 2025), and Modeling Microenvironment Trajectories (Sakalyan et al.). Optimal transport (Cuturi, 2013; Tong et al., 2023) is also widely used for computational biology (Klein et al., 2025) More related works are provided in the Appendix D.6

## 3 PROBLEM FORMULATION

Spatial transcriptomics technologies provide unprecedented opportunities to study biological systems by capturing gene expression profiles while preserving spatial location within tissue samples. Recent advancements, particularly in image-based spatial transcriptomics, offer high-resolution data that includes both cellular morphology and gene expression at a single-cell level (Janesick et al., 2023). This presents a unique challenge and opportunity to develop computational frameworks that can effectively integrate these rich, multimodal data sources to gain a deeper understanding of cellular states and interactions within their native spatial context.

We consider learning a unified, multi-scale representations that integrate cellular morphology and gene expression from spatial transcriptomic (Fig. 1). We begin with a cell-level dataset:

$$\mathcal{D}_{\text{cell}} = \{(\mathbf{C}_i, \mathbf{g}_i, s_i)\}_{i=1}^M, \tag{1}$$

where $\mathbf{C}_i \in \mathbb{R}^{H \times W \times 3}$ is the high-resolution cropped image of cell $i$, $\mathbf{g}_i \in \mathbb{R}^G$ is its gene-expression vector of the cell, and $s_i = (x_i, y_i) \in \mathbb{R}^2$ denotes its spatial coordinate. We learn a single embedding $\mathbf{z}_i^c$ for each cell $i$ that captures its morphology, transcriptome, and spatial context in one coherent vector via a model:

$$\mathbf{z}_i^c = \mathcal{F}_{cell}(\mathbf{C}_i, \mathbf{g}_i, \mathbf{s}_i) \in \mathbb{R}^D \tag{2}$$

Next, we group cells into non-overlapping spatial *niches* of the slides with a corresponding set of cells $\mathcal{C}_j$. For niche $j$, we add the gene-expression vectors of the cells $\mathbf{g}_j^n = \sum_{i \in \mathcal{C}_j} \mathbf{g}_i$, and encode its larger image patch $P_k^{\text{reg}}$ using a dedicated niche-level encoder $\mathcal{F}_{niche}$. Similarly at the tissue level, we group cells into larger non-overlapping spatial regions of the slides with a corresponding set of cells $\mathcal{C}_k$. We pass the data into a global transformer $\mathcal{F}_{tissue}$ to capture long-range dependencies across the entire slide. The niche and tissue level dataset and embeddings are:

$$\begin{cases} \mathcal{D}_{\text{niche}} = \{(\mathbf{N}_j, \mathbf{g}_j^n, s_j)\}_{j=1}^P, \\ \mathcal{D}_{\text{tissue}} = \{(\mathbf{T}_k, \mathbf{g}_k^t, s_k)\}_{k=1}^Q \end{cases}, \quad \begin{cases} \mathbf{z}_j^n = \mathcal{F}_{niche}(\mathbf{N}_j, \mathbf{g}_j^n, \mathbf{s}_j), \\ \mathbf{z}_k^t = \mathcal{F}_{tissue}(\mathbf{T}_k, \mathbf{g}_k^t, \mathbf{s}_k) \end{cases} \in \mathbb{R}^D, \tag{3}$$

To obtain the final unified embedding for each cell $i$, we attend from its cell-level representation to both the corresponding niche and the global tissue context:

$$\mathbf{z}_i = \mathcal{F}_{\text{fusion}}\Big( \underbrace{\mathbf{z}_i^c}_{\text{cell level}}, \underbrace{\mathbf{z}_j^n}_{\text{niche level}}, \underbrace{\mathbf{z}_k^t}_{\text{tissue level}} \Big) \in \mathbb{R}^D, \tag{4}$$

The resulting unified embedding $\mathbf{z}_i$, along with the intermediate scale-specific embeddings $\mathbf{z}_i^c$, $\mathbf{z}_i^n$, and $\mathbf{z}_i^t$, can be leveraged for a wide range of downstream biomedical tasks. These include: (i) *Morphology prediction*, using $\mathbf{z}_i$ as condition to synthesize realistic cell images via generative modules; (ii) *spatial identification*, through clustering or regional softmax over the embeddings; (iii) *cell type annotation*, by training an MLP classifier on $\mathbf{z}_i$; and (iv) *gene expression imputation*, by learning a regression model that maps $\mathbf{z}_i$ to its corresponding gene expression vector, etc

## 4 SPATIA MODEL

As shown in Fig. 2A, SPATIA is designed to learn comprehensive, multi-scale representations by integrating cellular morphology, gene expression, and spatial context from image-based spatial transcriptomics data. It operates hierarchically, first learning unified single-cell representations by fusing cell image and gene data, then refining these representations by modeling spatial relations within the tissue microenvironment.

### 4.1 UNIFIED SINGLE-CELL REPRESENTATION LEARNING

We aim to generate a unified embedding for each cell that captures synergistic information from its morphology and gene expression profile. We employ separate encoders for the image and gene expression modalities to extract initial feature representations for each cell $i$.

**Morphological Feature Extraction.** Each cropped and standardized cell image $\mathbf{C}i$ is processed by a ViT-based encoder $E$cell, which divides the image into patches and projects them into a sequence of visual tokens forming the cell matrix $\mathbf{X}_i^c$.

**Gene Expression Feature Extraction.** The gene expression vector $\mathbf{g}_i$ is encoded using the pre-trained scPRINT backbone (Kalfon et al., 2025), producing a token matrix $\mathbf{X}i^g \in \mathbb{R}^{N\text{gene} \times D}$ that captures gene-level dependencies and expression patterns.

**Multimodal Feature Fusion.** To integrate information from both modalities at the single-cell level, we employ a cross-attention mechanism, as depicted in Fig. 2. Specifically, we use the cell matrix $\mathbf{X}_i^c$ as the query sequence and the gene matrix $\mathbf{x}_i^g$ as the key and value sequences. The fusion module then produces the embedding $\mathbf{z}_i^c = \text{CrossAttn}(Q = \mathbf{X}_i^c, K = \mathbf{X}_i^g, V = \mathbf{X}_i^g)$, which aligns fine-grained morphological tokens with the transcriptomic tokens to obtain the single, unified representation for cell $i$, that encapsulates fused morphological and transcriptomic information.

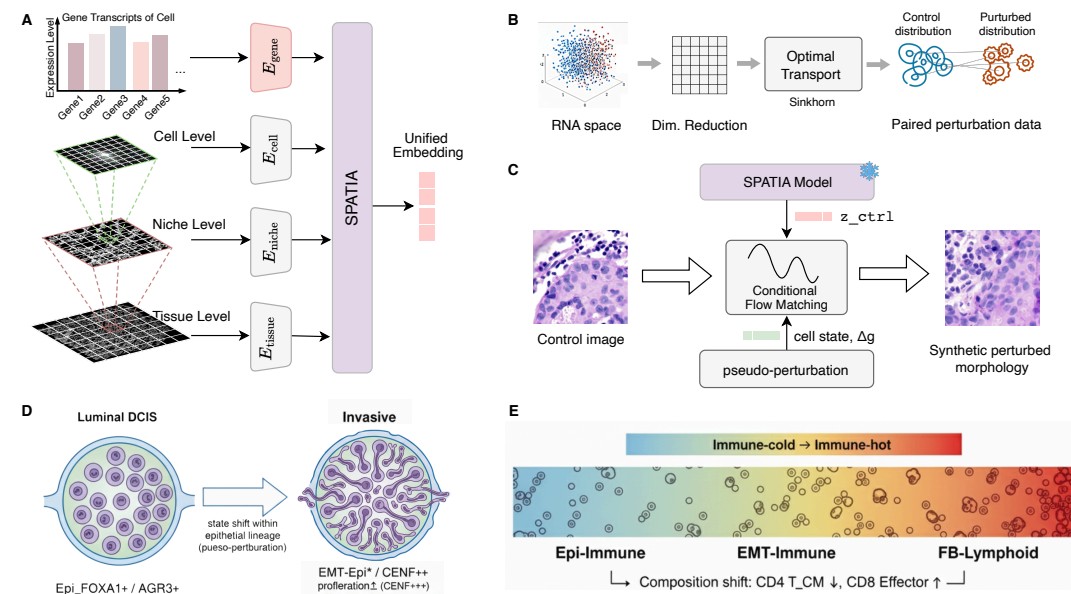

Figure 2: A) Overview of SPATIA. B) Processing control–target pairs with optimal transport. C) Our conditional contrastive flow matching approach for predicting cell morphology. D) The progression from luminal ductal carcinoma in situ to invasive carcinoma. E) Modeling the shift from an immune-cold tumor microenvironment to an immune-hot one.

## 4.2 HIERARCHICAL SPATIAL INTEGRATION VIA MULTI-LEVEL TRANSFORMERS

Building upon the unified cell representations derived earlier, SPATIA employs a multi-level learning approach with dedicated transformer modules to integrate spatial context and learn representations at progressively larger scales: the niche level and tissue (slide) level. This approach enables the model to capture cellular interactions within neighboring niches as well as in the global tissue area. Moreover, it allows the cell representation to obtain spatial-aware relational information.

**Niche Representation Learning.** We define a niche as a spatial region containing neighboring cells (Fig. 3), which characterizes local tissue structures such as tumor or immune niches, differing in both gene expression and cell-type composition. Inside each niche, we align cell IDs with gene expression and incorporate spatially dependent labels derived from expression similarity and proximity. This enables microenvironmental features that capture heterogeneity critical for cellular interactions.

Given a niche image $\mathbf{N}_j$ and cell set $\{c_i | c_i \in \mathcal{C}_j\}$, we extract morphological tokens $\mathbf{X}_j^n = E_{\text{niche}}(\mathbf{N}_j) \in \mathbb{R}^{N_{\text{patch}} \times D}$ and pool per-cell embeddings into a niche context vector $\bar{\mathbf{z}}_j = \frac{1}{|\mathcal{C}_j|} \sum_{i \in \mathcal{C}_j} \mathbf{z}_i^c \in \mathbb{R}^D$. We then fuse morphology and cell context via cross-attention:

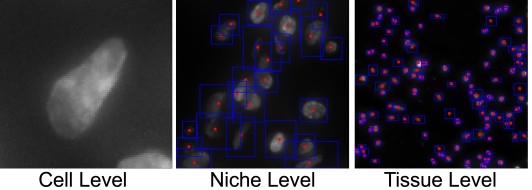

Figure 3: Three levels of whole slide image.

$$\mathbf{z}_j^n = \text{CrossAttn}(Q = \bar{\mathbf{z}}_j, K = \mathbf{X}_j^n, V = \mathbf{X}_j^n) \in \mathbb{R}^D, \tag{5}$$

producing a niche representation $\mathbf{z}_j^n$ that encodes local cell–cell interactions.

**Tissue Representation Learning.** At the tissue (slide) level, transformers aggregate information from all niches to capture global patterns and long-range interactions. The input sequence consists of niche embeddings $\{z_k^n\}_{k=1}^{N_{\text{regions}}}$ with positional encodings $pe_k$ indicating their grid location. Cross-attention integrates each niche with the global tissue context:

$$\mathbf{z}_k^t = \text{CrossAttn}(Q = \mathbf{z}_k^n, K = \mathbf{X}_k^t, V = \mathbf{X}_k^t) \in \mathbb{R}^D. \tag{6}$$

Unlike multi-instance learning (Gadermayr & Tschuchnig, 2024), which summarizes instances by pooling, SPATIA explicitly models spatial relationships and integrates information across multiple scales, from single cells to niches, and finally to the slide level, leveraging the strengths of transformers for context aggregation at each level.

We train the SPATIA using self-supervised objectives with paired multimodal data. We enforce consistency between the modalities and learn the fusion process by reconstructing the original data (Appendix B) from the unified embedding via a dual decoder.

### 4.3 SPATIALLY CONDITIONED MORPHOLOGY GENERATION

Cellular responses to perturbations depend strongly on the surrounding microenvironment, which governs exposure to signaling molecules and opportunities for cell-cell communication. Experimentally profiling morphological outcomes for all perturbations is infeasible, both because the perturbation space is enormous and because sequencing is destructive, preventing observation of the same cell before and after perturbation. These constraints motivate the need for in-silico models that can predict morphology under perturbations while accounting for spatial context. SPATIA addresses this by conditioning morphology generation on both intrinsic cell state and extrinsic spatial environment, capturing microenvironmental cues that prior models overlook.

**Weak Pair Construction.** We construct *weak control–target pairs* at the distribution level. Specifically, we match cells of the same type within spatially adjacent or niche-consistent regions, defining one group as the control (pre-perturbation) and the other as the target (post-perturbation). For instance, near-normal or low-malignancy epithelial cells can serve as controls, while invasive carcinoma cells of the same lineage are treated as perturbed targets (Fig. 2D). Similarly, T cells in immune-cold versus immune-hot regions may be paired (Fig. 2E), as can tumor cells of different molecular subtypes. Each pair is recorded with identifiers and metadata, $\{x_{\mathrm{ctrl\_id}}, x_{\mathrm{tgt\_id}}, \mathrm{state}_A, \mathrm{state}_B, \mathrm{niche}, \mathrm{cell\ type}\}$, yielding a structured dataset for training.

We formulate the pairing task as an *optimal transport* problem, under the principle that biologically similar cells should be matched preferentially based on their expression profiles (Fig. 2B). We implement this by solving for the entropy-regularized transport plan via the Sinkhorn-Knopp algorithm (Cuturi, 2013), which aligns the global distributions of control and target populations (Tong et al., 2023). This minimizes the aggregate transport cost defined by Euclidean distances in the reduced PCA expression space. Importantly, this pairing is performed in gene expression space rather than image space, avoiding trivial morphological matches and ensuring that pairs reflect underlying molecular state changes that drive morphology. Spatial proximity constraints further prevent mismatches arising from tissue heterogeneity.

**Pseudo-Perturbation Embedding.** Building on these weak pairs, we derive perturbation features that condition the generative model on state transitions. For each defined transition (e.g., normal → DCIS, DCIS → invasive; immune-cold → immune-hot), we encode state labels as ordered transition tokens and compute differential expression signatures $\Delta g$ between matched control and target cells within the same lineage and niche. The $\Delta g$ vectors are reduced to a low-dimensional representation via PCA and fused with the transition tokens through an MLP, yielding a compact perturbation embedding. A detailed pipeline is provided in Appendix D.

**Flow Matching for Control–to–Target Generation.** We propose a conditional contrastive flow matching approach for predicting cell morphology (Fig. 2C). Given a control image $x_{\mathrm{ctrl}}$ and a weakly matched target $x_{\mathrm{tgt}}$, we encode $\ell_0 = \mathrm{Enc}(x_{\mathrm{ctrl}})$, and $\ell_1 = \mathrm{Enc}(x_{\mathrm{tgt}})$ and define the linear bridge $\ell_t = (1 - t)\,\ell_0 + t\,\ell_1, t \sim \mathcal{U}(0, 1)$. A velocity network $v_\theta$ is conditioned on two signals: (i) the pseudo-perturbation embedding $e^{\mathrm{pert}}$ and (ii) the frozen SPATIA control embedding $z_{\mathrm{ctrl}}$ via FiLM modulation. At inference, we integrate the learned field starting at $\ell_0$ under the same conditions $(e^{\mathrm{pert}}, z_{\mathrm{ctrl}})$ to obtain a perturbed latent $\hat{\ell}$, then decode to a synthetic $\hat{x}_{\mathrm{tgt}}$ that preserves cell identity/context while expressing the morphological change.

Conditional FM can blur condition-specific signals when conditional distributions overlap, yielding trajectories that are insufficiently discriminative across conditions (Stoica et al., 2025). To address this, we add a contrastive term that discourages similar flows under different conditions. Concretely, for each training example we draw a negative $(x'_{\mathrm{ctrl}}, x'_{\mathrm{tgt}}, e^{\mathrm{pert}-}, z'_{\mathrm{ctrl}})$ with a different transition/niche condition and define $\ell'_0 = \mathrm{Enc}(x'_{\mathrm{ctrl}})$, $\ell'_1 = \mathrm{Enc}(x'_{\mathrm{tgt}})$. We then penalize alignment of the positive

flow to the negative target direction:

$$\mathcal{L}_{\text{cFM}}(\theta) = \mathcal{L}_{\text{FM}}(\theta) + \rho \, \mathbb{E}_{(x,x')} \, \mathbb{E}_{t \sim \mathcal{U}(0,1)} \Big[ \big\| v_\theta(\ell_t, t \mid e^{\text{pert}}, z_{\text{ctrl}}) - (\ell_1' - \ell_0') \big\|_2^2 \Big], \qquad (7)$$

with $\rho \in [0, 1)$ controlling contrastive strength and $\mathcal{L}_{\text{FM}}$ as the standard flow-matching loss. This pull-together within condition, push-apart across conditions objective preserves the controllability of conditional FM while encouraging condition-distinct flows, improving faithfulness to $e^{\text{pert}}$ and reducing averaged-out outputs.

## 5 EXPERIMENTS

### 5.1 DATASETS AND EXPERIMENTAL SETUP

**MIST Datasets.** MIST (Multi-scale dataset for Image-based Spatial Transcriptomics) dataset is assembled from 49 Xenium sources (Janesick et al., 2023) spanning 17 tissue types, 49 donors, and 12 disease states. MIST comprises three nested scales: 1) **MIST-C**: 17M single cell-gene pairs; 2) **MIST-N**: 1M niche-gene pairs; 3) **MIST-T**: 10K tissue-gene entries. These splits enable precise mapping of cell morphology to transcriptomics at cellular, regional, and whole-slide levels, supporting multimodal representation learning across diverse biological contexts. We first load the full-resolution tissue image ($0.2125\mu$m/px) and compute a maximum-intensity projection over z. The resulting 2D image is normalized to 8-bit $[0, 255]$. We use the cell boundary file to extract individual cell images. For each cell, we compute the minimal square region that fully contains the cell and crops the image accordingly. Each cell-gene example consists of this uint8 image patch and the corresponding per-cell transcript vector for a single gene, serialized into LMDB for efficient training (MIST-C). To form MIST-N, we tile each slide into a grid of non-overlapping $256 \times 256$ px niches, assign cells to their containing patch, and pool gene vectors within each niche. Each niche entry therefore includes the regional image patch and its aggregated gene profile. Finally, MIST-T summarizes each slide by its set of niche embeddings and positional metadata, with a size of $1024 \times 1024$, enabling tissue-level tasks such as global composition prediction and cross-slide transfer. The full dataset statistics are present in Appendix A.

**Baselines.** We benchmark SPATIA against various models, including CellFlux (Zhang et al., 2025) and MorphDiff (Wang et al., 2025b) for cell morphology prediction; UNI (Chen et al., 2024b), GigaPath (Xu et al., 2024), Hibou (Nechaev et al., 2024), CLIP, PLIP (Huang et al., 2023), CONCH (Lu et al., 2023), CTransPath (Wang et al., 2022), UNIv1.5 (Chen et al., 2024b) and H-Optimus-0 (Saillard et al., 2024) for biomarker status prediction and gene expression prediction; as well as single-cell models: Geneformer (Theodoris et al., 2023), scGPT (Cui et al., 2023), CellTypist (Domínguez Conde et al., 2022), scBERT (Yang et al., 2022), and CellPLM (Yang et al., 2022) for cell annotation & clustering.

**Experimental Setup and Implementation.** We evaluate SPATIA on control–to–target generation of cell morphology. Additionally, we evaluate SPATIA across four groups of tasks: cell annotation, clustering, gene expression prediction, and biomarker status prediction. We mainly followed the downstream evaluation settings from (Wen et al., 2023) and (Jaume et al., 2024) Detailed training settings and model configurations are provided in Appendix C.

### 5.2 CONTROL-TO-TARGET GENERATION OF CELL MORPHOLOGY

We perform two biological transitions: 1) tumor progression from ductal carcinoma in situ (DCIS) to invasive carcinoma within luminal epithelial cells. 2) Immune remodeling in which T cells are generated under immune-cold versus immune-hot microenvironments.

Evaluation proceeds along two axes. Image fidelity is assessed using Frechet Inception Distance (FID) (Heusel et al., 2017) and Kernel Inception Distance (KID) (Bińkowski et al., 2018), providing standard measures of generative realism. Mor-

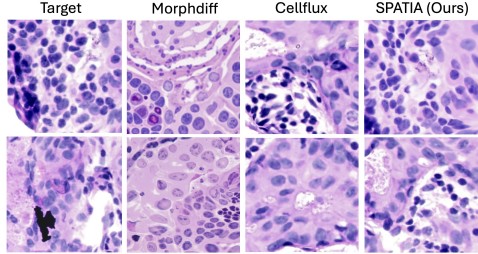

Figure 4: Comparison of generated cell morphology change images with the target image

phological correctness is assessed using CellProfiler-derived features (Carpenter et al., 2006).For each feature, generated and real distributions in the target state are compared using statistical distances such as the Kolmogorov–Smirnov (KS) statistic (Massey Jr, 1951) and the Wasserstein distance (Panaretos & Zemel, 2019). This dual evaluation ensures that generated images are not only visually realistic but also biologically faithful.

Our results in Table 1 highlight both the fidelity and the biological validity of our generative framework. Compared to CellFlux and MorphDiff, our method achieves lower FID and KID scores, indicating more realistic synthesis of cell images. At the same time, higher Wasserstein correlations and KS statistics demonstrate that generated morphologies more faithfully reproduce the distribution of CellProfiler-derived features in the target states. We also show visualization comparisons of generated samples in Fig. 4.

Table 1: Conditional generation of niche level cell morphology using FM in SPATIA

| Task | Image fidelity | | Morphology correctness | |
|------|------|------|------|------|
| | FID $\downarrow$ | KID $\downarrow$ | Wass. Corr. $\uparrow$ | KS $\uparrow$ |
| CellFlux | 64.1 | 2.31 | 0.87 | 0.57 |
| MorphDiff | 70.5 | 2.52 | 0.83 | 0.54 |
| SPATIA | 59.2 | 2.04 | 0.92 | 0.62 |

## 5.3 BIOMARKER STATUS PREDICTION

We compare SPATIA against six models on invasive breast cancer dataset in Table 2, evaluating ER, PR, and HER2 status from WSIs in the BCNB dataset. We follow the settings in (Chen et al., 2022) to embed the data into existing pathology models using modality-specific encoders. For example, we implement the image patches using a pretrained model (e.g., CONCH) and the expression data using a 3-layer MLP.

The modality-specific embeddings are then aligned using a contrastive objective, i.e., InfoNCE loss, by fine-tuning the image encoder and training the expression encoder from scratch. SPATIA consistently achieves the highest AUC and balanced accuracy across all three markers.

1) SPATIA attains an AUC of 0.902 and balanced accuracy of 0.785 for ER, improving over the prior best UNI by +0.011 AUC and +0.010 Bal.acc. 2) Our model

Table 2: Receptor–status prediction evaluation on BCNB

| Model | ER | | PR | | HER2 | |
|-------|------|------|------|------|------|------|
| | AUC | Bal.acc. | AUC | Bal.acc. | AUC | Bal.acc. |
| UNI | 0.891 | 0.775 | 0.820 | 0.712 | 0.732 | 0.641 |
| GigaPath | 0.841 | 0.765 | 0.803 | 0.696 | 0.721 | 0.635 |
| Hibou | 0.832 | 0.754 | 0.801 | 0.694 | 0.705 | 0.630 |
| CLIP | 0.652 | 0.537 | 0.618 | 0.502 | 0.514 | 0.438 |
| PLIP | 0.712 | 0.603 | 0.695 | 0.587 | 0.611 | 0.524 |
| CONCH | 0.881 | 0.745 | 0.810 | 0.698 | 0.715 | 0.624 |
| SPATIA | **0.902** | **0.785** | **0.825** | **0.730** | **0.744** | **0.643** |

reaches AUC 0.825 and Bal.acc 0.731, compared to UNI's 0.820 AUC and 0.712 Bal.acc, a gain of +0.005 AUC and +0.019 Bal.acc. 3) HER2: SPATIA records AUC 0.744 and Bal.acc 0.643, outperforming UNI by +0.012 AUC and +0.002 Bal.acc. These demonstrate that integrating multiscale spatial context with yields enhanced capacity to capture morphological and molecular signals.

## 5.4 CELL ANNOTATION & CLUSTERING

We follow the settings in (Wen et al., 2023) and use Multiple Sclerosis (MS) dataset (Schirmer et al., 2019) to evaluate cell annotation performance and scRNAseq data (Li et al., 2020). Results are shown in Tab. 3. We report F1 and Precision scores for annotation task; ARI and NMI scores for clustering task. These re-

Table 3: Cell annotation and clustering results.

| Method | Annotation | | Method | Clustering | |
|--------|------|------|--------|------|------|
| | F1 ($\uparrow$) | Precision ($\uparrow$) | | ARI ($\uparrow$) | NMI ($\uparrow$) |
| scGPT | 0.703 | 0.729 | PCA | 0.843 | 0.812 |
| CellPLM | 0.709 | 0.702 | CellPLM | 0.867 | 0.823 |
| scBERT | 0.599 | 0.604 | scGPT | 0.856 | 0.828 |
| CellTypist | 0.667 | 0.693 | Geneformer | 0.461 | 0.586 |
| SPATIA | 0.725 | 0.734 | SPATIA | 0.870 | 0.831 |

sults highlight SPATIA's ability on supervised and unsupervised single-cell analysis tasks compared to existing methods.

## 5.5 GENE EXPRESSION PREDICTION FROM IMAGES

We use HEST-Bench (Chen et al., 2022) to evaluate SPATIA for gene expression prediction task. Fig. 5 reports Pearson correlation coefficients (PCC) for the top 50 highly variable genes on five

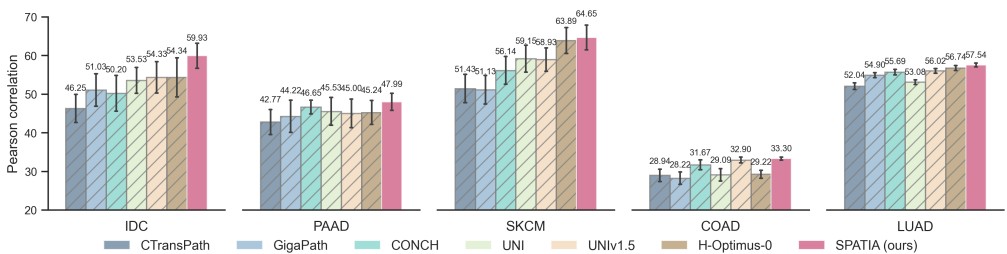

Figure 5: Gene expression prediction.

cancer cohorts (IDC, PAAD, SKCM, COAD, LUAD). We train a regression model to map model-specific patch embeddings to the log1p-normalized expression of the top 50 highly variable genes. We use XGBoost (Chen & Guestrin, 2016) regression model with 100 estimators and a maximum depth of 3. These consistent gains across diverse tissue types demonstrate that SPATIA yields embeddings that more accurately capture gene-image relationships than existing single or dual modal architectures.

## 6 ABLATION & ANALYSIS

**Model Component Effectiveness.** Table 4 reports an ablation on cell-type classification, starting from the cell-level backbone and incrementally adding: (i) reconstruction loss, (ii) multi-level hierarchy, (iii) cross-attention fusion. Adding the multi-level hierarchy produces the largest single improvement (0.27 loss ↓), underscoring the value of aggregation across scales. Collectively, these results show that each component meaningfully enhances our multi-scale representation.

Table 4: Sub-module effectiveness evaluation of SPATIA

| Method | Loss (↓) | Accuracy (↑) |
|---|---|---|
| Cell level only | 0.405 | 0.93 |
| + MAE loss | 0.396 | 0.94 |
| + Multi-level | 0.369 | 0.97 |
| + Fusion | 0.361 | 0.98 |

**Pairing Error Analysis.** To assess the robustness of the conditional flow generation module to imperfect OT-based controlperturbed matching, we conducted a pairing-noise ablation.

While our OT procedure incorporates lineage consistency and a spatial-adjacency penalty to discourage implausible matches, the flow model itself is designed to learn distributional perturbation directions rather than exact one-to-one trajectories. We therefore randomly corrupted 1020% of OT pairs by swapping per-

Table 5: Conditional generation Evaluation

| Noise Level | Image fidelity | | Morphology correctness | |
|---|---|---|---|---|
| | FID ↓ | KID ↓ | Wass. Corr. ↑ | KS ↑ |
| 0% | 59.2 | 2.04 | 0.92 | 0.62 |
| 10% | 61.0 | 2.12 | 0.90 | 0.60 |
| 20% | 63.8 | 2.25 | 0.88 | 0.58 |

turbed targets within the same slide and retrained the flow module under identical settings. As shown in Tab. 5, performance degrades smoothly with increasing noise (e.g., FID: 59.2 61.0 63.8), confirming that SPATIA remains stable under moderate pairing errors and does not rely on brittle correspondences during conditional generation.

**Multi-level Effectiveness.** To directly test whether the conditional flow is exploiting dataset co-occurrence rather than genuine spatial conditioning, we train a cell-only variant of SPATIA that retains the same cell-level image and gene encoders and the conditional flow module, but removes all niche/tissue embeddings and multi-scale spatial fusion. This ablation isolates the effect of spatial context while keeping architectural capacity comparable. The results are shown in Tab. 6

Table 6: Multi-level Effectiveness

| Noise Level | Image fidelity | | Morphology correctness | |
|---|---|---|---|---|
| | FID ↓ | KID ↓ | Wass. Corr. ↑ | KS ↑ |
| SPATIA cell only | 64.3 | 2.44 | 0.87 | 0.56 |
| SPATIA | 59.2 | 2.04 | 0.92 | 0.62 |

## 7 CONCLUSION

SPATIA is a multi-resolution model that integrates cellular morphology, spatial context, and gene expression for spatial transcriptomics. The model addresses a critical gap in existing approaches,

which often treat these modalities in isolation and fail to capture the structured dependencies across biological scales. SPATIA achieves strong performance on a range of predictive and generative benchmarks, including cell type classification, gene expression imputation, spatial clustering, and conditional morphology generation. The hierarchical attention modules model local cellcell interactions and long-range dependencies, and as we show, self-supervised objectives, including cross-modal reconstruction and flow-based image-to-image generation. The model is trained and evaluated on MIST, a large multi-scale dataset assembled from 49 image-based spatial transcriptomics samples across 17 tissue types and 12 disease contexts. MIST provides one-to-one mappings between image patches and transcriptomic profiles at single-cell, niche, and tissue levels. SPATIA provides a foundation for modeling spatial omics with fine-grained resolution. Future work will explore extending the framework to additional spatial omics modalities, training on more single-cell data, integrating temporal dynamics, and scaling to larger cohorts for clinical applications.

## ETHICS STATEMENT

This study uses publicly available spatial transcriptomics and imaging datasets (e.g., Xenium, Visium) that were collected and released under appropriate institutional and ethical approvals, and no new data involving human or animal subjects were generated. Our work focuses on developing machine learning methods for morphology generation under pseudo-perturbations, and does not involve personally identifiable information or sensitive data. All models are trained and evaluated on de-identified data, and no clinical or therapeutic claims are made. We acknowledge that generative models can be misused if applied beyond their intended scientific scope; to mitigate this risk, we restrict our analysis to academic research settings and provide clear documentation of our assumptions and limitations. We have carefully adhered to the ICLR Code of Ethics throughout the preparation and submission of this work.

## REPRODUCIBILITY STATEMENT

We have taken several steps to ensure the reproducibility of our work. All model details, including the flow-matching architecture, conditioning strategy, and loss functions, are described in Section 4 and Appendix B. The procedures for constructing weak controltarget pairs and generating pseudo-perturbation embeddings are outlined in Section 4.4, with further data preprocessing details provided in the supplementary materials. Hyperparameters, training configurations, and ablations are fully reported in the appendix. Evaluation protocols, including FID, KID, and CellProfiler-based morphology metrics (KS and Wasserstein distances), are defined in Section 5. To facilitate reproducibility, we provide anonymized source code and data processing scripts as part of the supplementary material at submission.

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

## A  MIST DATASET

**Dataset Statistics.** Imaging-based spatial transcriptomics technologies allow us to explore spatial gene expression profiles at the cellular level. Xenium is a high-resolution, imaging-based in situ spatial profiling technology from 10x Genomics that allows for simultaneous expression analysis of RNA targets within the same tissue section. This assay can identify the location of target transcripts within the tissue, providing a single cell resolution map of expression patterns of all genes that are included in the selected probe panel. Our MIST dataset is assembled from 49 Xenium sources Janesick et al. (2023) spanning 17 tissue types, 49 donors, and 12 disease states, including: healthy (7) Breast cancer (5) Lung cancer (4) Adenocarcinoma (3) Ovarian cancer (2) cancer (melanoma) (2) Cervical cancer (1) Colorectal cancer (1) Invasive Ductal Carcinoma (1) Melanoma (1) acute lymphoid leukemia (1) cancer (1). The total dataset contains 17,515,676 total cells and 6,000 unique genes. Dataset Statistics is shown in Tab. 7 and Tab. 8.

**Data Processing.** We address varying cell sizes by first computing a bounding box for each cell and determining a global scale factor from the largest bounding box in the slide. All cells are resized using this single scale, which preserves biologically meaningful variation in absolute cell size. For each cell, the cropped patch is resized with the global scale and then padded to $256 \times 256$, ensuring a fixed input dimension while keeping only that cell in the image. Padding prevents pixels from neighboring cells, which correspond to different expression vectors from being incorporated, avoiding modality mismatch. Additionally, Xenium provides high-quality cell contours, which we retain to preserve exact spatial size information even after resizing and padding.

In MIST, niches are defined using a non-overlapping $256 \times 256$ px fixed grid applied uniformly across the slide (Xenium resolution: 0.2125 m/px). All cells whose centroids fall within a grid tile are grouped into the same niche. For each niche, we aggregate the gene expression vectors of its constituent cells (using the pooled representation described earlier) and extract the corresponding regional image patch. This choice follows widely adopted patch-based strategies in spatial transcriptomics and computational pathology (Navidi et al., 2025; Huang et al., 2025b; Fu et al., 2025b; Huang et al., 2025b). We also empirically validated that the chosen niche size is biologically reasonable. A $256 \times 256$ px region typically contains around 10-30 cells, depending on tissue density, which aligns with common definitions of microenvironments such as tumor margins, lymphocytic aggregates, and stromal niches in pathology. We visualize this distribution in Fig. 3. At the tissue level, we group $4 \times 4$ neighboring niches into a $1024 \times 1024$ px region, enabling the model to capture coarse-scale patterns such as tumor invasion fronts and broad architectural organization. This multi-level design allows SPATIA to model both local neighborhood interactions and larger-scale spatial structure.

**Batch Effect Discussion.** To assess potential batch effects in the MIST atlas, we analyzed all Xenium datasets by constructing a common gene space (70,611 shared genes), sampling 2,000 cells from each dataset, and performing joint PCA followed by UMAP. Silhouette Scores computed on the PCA embeddings show low cluster separation by donor, and UMAP visualizations confirm that cells organize primarily by biological identity rather than by dataset source. Results are shown in Tab. 6. These findings suggest that technical batch variation is modest relative to biological variation in this setting. Our design is consistent with recent spatial-omics foundation models such as scGPT-spatial, which use principled normalization plus large-scale pretraining to implicitly mitigate batch effects across Visium and Xenium slides, and with visualomics foundation models that rely on normalization and cross-modal training rather than bespoke correction for each dataset. We also note that the spatial transcriptomics community is still actively debating how to handle batch and library-size effects, and over-normalization or aggressive batch correction can distort spatial domains rather than improve them.

**Potential Information Leakage Discussion.** Since cell-level embeddings attend to niche/tissue features, there is a risk of information leakage across scales. To prevent this, our pretraining is entirely self-supervised and contains no perturbation labels or signatures, so no nicheperturbation leakage can occur. Most downstream tasks are single-level and do not combine niche-level perturbation signals with cell-level labels. For tasks where both cell and niche representations are used, the model receives both modalities as explicit inputs, not as labels, so niche correlations do not generate shortcut pathways.

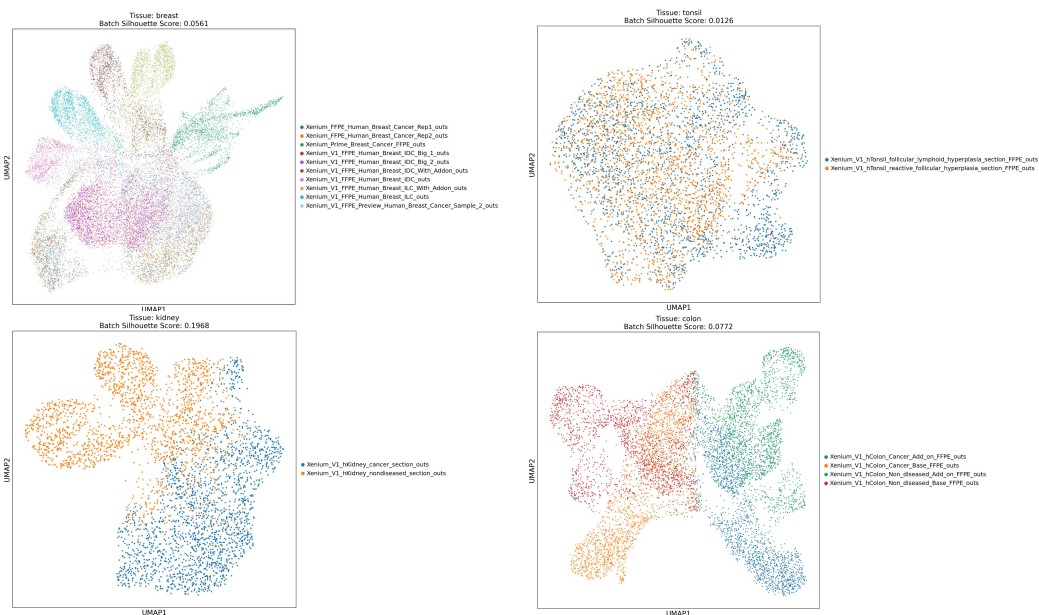

Figure 6: Visualization of Batch Effect among donors

Additionally, all evaluations use donor-disjoint splits, meaning that all modalities (morphology, expression, spatial context) from a donor appear exclusively in either the training set or the test set. Because donor identity is the dominant source of morphological variation, this prevents tissue-level morphology from leaking into the prediction task. Moreover, the performance gains persist even when using single-modality ablations (only morphology or only expression), confirming that improvements are driven by the learned multimodal representations rather than unintended cross-slide or cross-donor leakage.

## B    SELF-SUPERVISED TRAINING OBJECTIVES

We train the SPATIA using self-supervised objectives with paired multimodal data. We enforce consistency between the modalities and learn the fusion process by reconstructing the original data from the unified embedding via a dual decoder.

**Image Reconstruction.** An image decoder $D_{\text{cell}}$ takes the unified embedding $\mathbf{z}_i^c$ as input and aims to reconstruct the original cell image patch $\hat{\mathbf{C}}_i = D_{\text{cell}}(\mathbf{z}_i^c)$. The reconstruction loss $\mathcal{L}_{\text{img\_recon}}$ is typically the MAE (He et al., 2021) between the reconstructed and original image pixels. $\mathcal{L}_{\text{img\_recon}} = \frac{1}{N} \sum_{i=1}^{N} \|\hat{\mathbf{C}}_i - \mathbf{C}_i\|_2^2$

**Gene Reconstruction.** A gene decoder $D_{\text{gene}}$ takes the unified embedding $\mathbf{z}_i^c$ (or $\mathbf{z}_j^n$, $\mathbf{z}_k^t$) and aims to reconstruct the original gene expression profile $\mathbf{g}_i$. Using a set of learnable gene query embeddings $\{q_g\}_{g=1}^{N_{\text{gene}}}$, the decoder can attend to the unified cell embedding $\mathbf{z}_i^c$ (acting as memory) to predict the gene embeddings $\hat{\mathbf{g}}_i = D_{\text{gene}}(\mathbf{z}_i^c)$. We use masked language modeling (Devlin et al., 2019) for the reconstruction loss.

The overall loss function combines the selected objectives $\mathcal{L}_{\text{total}} = \lambda_{\text{cell}}\mathcal{L}_{\text{img\_recon}} + \lambda_{\text{gene}}\mathcal{L}_{\text{gene\_recon}}$, weighted by hyperparameters $\lambda_{\text{cell}}$ and $\lambda_{\text{gene}}$.

## C    TRAINING & IMPLEMENTATION DETAILS

**Training Details.** The hierarchical modules ($\mathcal{F}_{cell}$, $\mathcal{F}_{niche}$, $\mathcal{F}_{tissue}$) are trained concurrently with the corresponding dataset using primary reconstruction losses (MAE) computed on the corresponding embeddings ($\mathbf{z}_i^c$, $\mathbf{z}_i^n$, $\mathbf{z}_i^t$). We adopt a hierarchical and localized batching strategy, which keeps

Table 7: Xenium Datasets gathered from 10x Genomics (part 1).

| Collection name | Tissue | Disease | Num. cells | Num. genes |
|---|---|---|---|---|
| Xenium_Preview_Human_Lung_Cancer_With_Add_on_2_FFPE | lung | cancer | 531,165 | 392 |
| Xenium_Preview_Human_Non_diseased_Lung_With_Add_on_FFPE | lung | healthy | 295,883 | 392 |
| Xenium_Prime_Breast_Cancer_FFPE | breast | cancer | 699,110 | 5101 |
| Xenium_Prime_Cervical_Cancer_FFPE | cervical | cancer | 840,387 | 5101 |
| Xenium_Prime_Human_Lung_Cancer_FFPE | lung | cancer | 278,328 | 5001 |
| Xenium_Prime_Human_Lymph_Node_Reactive_FFPE | lymph node | reactive hyperplasia | 708,983 | 4624 |
| Xenium_Prime_Human_Ovary_FF | ovary | adenocarcinoma | 1,157,659 | 5001 |
| Xenium_Prime_Human_Prostate_FFPE | prostate | adenocarcinoma | 193,000 | 5006 |
| Xenium_Prime_Human_Skin_FFPE | skin | melanoma | 112,551 | 5006 |
| Xenium_Prime_Ovarian_Cancer_FFPE | ovary | cancer | 407,124 | 5101 |
| Xenium_V1_FFPE_Human_Brain_Alzheimers_With_Addon | brain | alzheimers | 44,955 | 354 |
| Xenium_V1_FFPE_Human_Brain_Glioblastoma_With_Addon | brain | glioblastoma | 40,887 | 319 |
| Xenium_V1_FFPE_Human_Brain_Healthy_With_Addon | brain | healthy | 24,406 | 319 |
| Xenium_V1_FFPE_Human_Breast_IDC_Big_1 | breast | invasive ductal carcinoma | 892,966 | 280 |
| Xenium_V1_FFPE_Human_Breast_IDC_Big_2 | breast | invasive ductal carcinoma | 885,523 | 280 |
| Xenium_V1_FFPE_Human_Breast_IDC_With_Addon | breast | invasive ductal carcinoma | 576,963 | 380 |
| Xenium_V1_FFPE_Human_Breast_IDC | breast | invasive ductal carcinoma | 574,852 | 280 |
| Xenium_V1_FFPE_Human_Breast_ILC_With_Addon | breast | invasive lobular carcinoma | 365,604 | 380 |
| Xenium_V1_FFPE_Human_Breast_ILC | breast | invasive lobular carcinoma | 356,746 | 280 |
| Xenium_V1_Human_Brain_GBM_FFPE | brain | glioblastoma | 816,769 | 480 |
| Xenium_V1_Human_Colorectal_Cancer_Addon_FFPE | colorectal | cancer | 388,175 | 480 |
| Xenium_V1_Human_Ductal_Adenocarcinoma_FFPE | pancreas | ductal adenocarcinoma | 235,099 | 380 |
| Xenium_V1_Human_Lung_Cancer_Addon_FFPE | lung | cancer | 161,000 | 480 |
| Xenium_V1_Human_Lung_Cancer_FFPE | lung | cancer | 278,659 | 289 |
| Xenium_V1_Human_Ovarian_Cancer_Addon_FFPE | ovary | cancer | 247,636 | 480 |
| Xenium_V1_hBoneMarrow_acute_lymphoid_leukemia_section | bone marrow | acute lymphoid leukemia | 225,906 | 477 |
| Xenium_V1_hBoneMarrow_nondiseased_section | bone marrow | healthy | 84,518 | 477 |
| Xenium_V1_hBone_nondiseased_section | bone | healthy | 33,801 | 477 |
| Xenium_V1_hColon_Cancer_Add_on_FFPE | colon | cancer | 587,115 | 425 |
| Xenium_V1_hColon_Cancer_Base_FFPE | colon | cancer | 647,524 | 325 |
| Xenium_V1_hColon_Non_diseased_Add_on_FFPE | colon | healthy | 275,822 | 425 |
| Xenium_V1_hColon_Non_diseased_Base_FFPE | colon | healthy | 270,984 | 325 |
| Xenium_V1_hHeart_nondiseased_section_FFPE | heart | healthy | 26,366 | 377 |
| Xenium_V1_hKidney_cancer_section | kidney | cancer | 56,510 | 377 |
| Xenium_V1_hKidney_nondiseased_section | kidney | healthy | 97,560 | 377 |

sequence lengths bounded and independent of the total number of cells in a slide. The cell encoder is pretrained independently using individual (image, expression) pairs. A training batch contains a fixed B number of sampled cells, and attention is computed only within this batch, not across the full slide. For each sampled cell, its 256×256 px niche is extracted and encoded as one niche token (10-30 neighboring cells). The niche encoder is pretrained separately and does not process the entire tissue at once. The resulting complexity for three levels is $O((3B)^2)$, ensuring feasibility regardless of slide size. Pretraining each level individually and fine-tuning jointly over localized patches avoids any quadratic explosion and makes SPATIA scalable to slides with hundreds of thousands of cells. We use the AdamW optimizer Kingma & Ba (2017) with a learning rate of 1e-3. Regarding downstream tasks, we follow the settings from CellPLM and HEST-1k. Specifically, For Biomarker Status Prediction Tasks, we fine-tune the image encoder and train the expression encoder from scratch. We use a base learning rate of $10^{-4}$ for the image encoder and $10^{-3}$ gene expression encoder. Only the last 3 layers of the model were fine-tuned, with a layer-wise learning decay rate of 0.7. For Gene Expression Prediction Tasks, we utilize an XGBoost regression model with 100 estimators and a maximum depth of 3. We evaluate 3-4 seeds, and the standard deviation is around 0.05

Table 8: Xenium Datasets gathered from 10x Genomics (part 2).

| Collection name | Tissue | Disease | Num. cells | Num. genes |
|---|---|---|---|---|
| Xenium_V1_hLiver_cancer_section_FFPE | liver | cancer | 162,628 | 474 |
| Xenium_V1_hLiver_nondiseased_section_FFPE | liver | healthy | 239,271 | 377 |
| Xenium_V1_hLung_cancer_section | lung | cancer | 150,365 | 377 |
| Xenium_V1_hLymphNode_nondiseased_section | lymph node | healthy | 377,985 | 377 |
| Xenium_V1_hPancreas_Cancer_Add_on_FFPE | pancreas | cancer | 190,965 | 474 |
| Xenium_V1_hPancreas_nondiseased_section | pancreas | healthy | 103,901 | 377 |
| Xenium_V1_hSkin_Melanoma_Base_FFPE | skin | melanoma | 106,980 | 282 |
| Xenium_V1_hSkin_nondiseased_section_1_FFPE | skin | healthy | 68,476 | 377 |
| Xenium_V1_hSkin_nondiseased_section_2_FFPE | skin | healthy | 90,106 | 377 |
| Xenium_V1_hTonsil_follicular_lymphoid_hyperplasia_section_FFPE | tonsil | follicular lymphoid hyperplasia | 864,388 | 377 |
| Xenium_V1_hTonsil_reactive_follicular_hyperplasia_section_FFPE | tonsil | reactive follicular hyperplasia | 1,349,620 | 377 |
| Xenium_V1_humanLung_Cancer_FFPE | lung | cancer | 162,254 | 377 |
| Xenium_V1_human_Pancreas_FFPE | pancreas | cancer | 140,702 | 377 |
| Xeniumranger_V1_hSkin_Melanoma_Add_on_FFPE | skin | melanoma | 87,499 | 382 |

**Model Architecture.** Our model architecture, based on `scPrint`, has core components including: a `GeneEncoder` for processing gene expression data, which contains an embedding layer (`Embedding`) and a continuous value encoder (`ContinuousValueEncoder`); an image processing module based on `ViTMAEForPreTraining`, comprising a 12-layer ViT encoder (`ViTMAEEncoder`) and an 8-layer ViT decoder (`ViTMAEDecoder`); and an 8-layer `FlashTransformerEncoder` as the main sequence transformer.

The model also integrates multiple `FusionLayers` for multimodal feature fusion, an `ExprDecoder` for gene expression reconstruction, and multiple `ClsDecoders` for downstream classification tasks. Key hyperparameters are summarized in Tab. 10.

Pretrained weights are essential for SPATIAs performance. The scPRINT gene encoder is pretrained on millions of scRNA-seq cells and is specifically designed to denoise expression, correct batch effects, and infer genegene interactions; training a gene encoder of similar scale from scratch on MIST is not feasible and leads to substantial performance degradation. Likewise, the pretrained ViT image encoder provides strong morphology priors that significantly improve single-cell feature quality. To quantify this, we compared SPATIA with (i) pretrained vision encoders and (ii) the same architectures trained from scratch (random initialization), while keeping the gene encoder fixed (as scPRINT is currently one of the few large pretrained models for scRNA-seq).

**Design Selection.** In niche level, the expression vectors are summed across cells. The summation operation is only applied at the niche level to obtain a coarse regional representation, similar to how pseudo-spots are constructed by aggregating single-cell expression within fixed grid regions in prior works (Liu et al., 2023; Mason et al., 2024; Hao et al., 2024b). This aggregation is not used for any cell-level task (Tab. 2 and Tab. 3 of the manuscript). All cell-level modeling and multimodal fusion in SPATIA are performed via cross-attention, which is fully non-linear and learns context dependent relationships between morphology and gene expression features.

**Computation Analysis.** We profiled SPATIA on a full-scale training run using 4 NVIDIA H100 (80GB) GPUs for 25,000 steps.

Table 9 summarizes the key system statistics. The model requires 67 GB of VRAM per device (78.7% peak utilization), confirming that the full architecture fits comfortably within a single high-end GPU without model parallelism. During training, GPU utilization reaches 97%, with an average power draw of 436 W per device. The largest checkpoint completes in approximately 30 hours, while inference runs at low latency, making SPATIA practical for both research and downstream biological workflows.

Table 9: Computation profile of SPATIA during full-scale training.

| Metric | Value | Notes |
|---|---|---|
| GPU Hardware | $4 \times$ NVIDIA H100 (80GB) | Full training run |
| Training Steps | 25,000 | Standard configuration |
| VRAM Usage | 67 GB/device | 78.7% peak utilization |
| GPU Utilization | 97% peak | Stable during training |
| Power Consumption | 436 W avg. | Per GPU |
| Training Time | $\sim$30 hours | Per largest checkpoint |
| Inference Time | Low latency | Suitable for deployment |

Table 10: Model Hyperparameters for SPATIA

| Component | Parameter | Value |
|---|---|---|
| *Gene Processing Module* | | |
| Gene Encoder | Embedding Dimension | 256 |
| | Vocabulary Size (Genes) | 23122 |
| Expression Encoder | Output Dimension | 256 |
| | Dropout | 0.1 |
| *Core Transformer* | | |
| Flash Transformer Enc. | Number of Blocks | 8 |
| | Hidden Size (d_model) | 256 |
| | MLP Intermediate Size | 1024 |
| | Dropout | 0.1 |
| *Image Processing Module (ViTMAE)* | | |
| ViT Encoder | Hidden Size | 768 |
| | Number of Layers | 12 |
| | Patch Size | 16x16 |
| | MLP Intermediate Size | 3072 |
| ViT Decoder | Hidden Size | 512 |
| | Number of Layers | 8 |
| | MLP Intermediate Size | 2048 |
| *Fusion Layers* | | |
| Image Fusion Layer | Dimension | 768 |
| | Dropout | 0.1 |
| Expression Fusion Layer | Dimension | 256 |
| | Dropout | 0.1 |
| *Output Decoders* | | |
| Expression Decoder | Hidden Dimension | 256 |
| | Dropout | 0.1 |

# D    PERTURBATION PAIRING FOR GENERATION OF SPATIAL TRANSCRIPTOMIC CELL PHENOTYPES

To construct perturbation pairs from spatial transcriptomic data for training control-to-target image generation models, we use optimal transport to align cells across states, creating paired datasets that reflect genuine cellular transitions and allow the simulation of tumor progression and immune infiltration through morphological changes.

### D.1 PROBLEM FORMULATION

Given a spatial transcriptomic dataset $\mathcal{D} = \{(x_i, g_i, s_i, m_i)\}_{i=1}^N$ where $x_i \in \mathbb{R}^{H \times W \times 3}$ represents the cell image, $g_i \in \mathbb{R}^G$ denotes the gene expression profile, $s_i$ indicates the cell state annotation, and $m_i$ represents the spatial microenvironment (niche), we aim to construct a paired dataset $\mathcal{P} = \{(x_i^c, x_j^t, g_i^c, g_j^t, \Delta g_{ij}, \tau_{ij})\}$ for training perturbation-conditioned generative models.

Here $(x_i^c, g_i^c)$ represents a *control* cell and $(x_j^t, g_j^t)$ a *target* cell, with $\Delta g_{ij} = g_j^t - g_i^c$ denoting the differential gene expression signature and $\tau_{ij}$ indicating the type of biological transition. The key challenge is to identify controltarget pairs that reflect authentic perturbation responses rather than arbitrary state differences.

**Transition Design.** We focus on two major perturbation axes (tasks) derived from domain knowledge: The tumor progression axis ($\mathcal{T}_{tumor}$) encompasses cellular transitions that collectively model the progression from luminal ductal carcinoma in situ (DCIS) to invasive carcinoma. First, we model epithelial-mesenchymal transition (EMT) through the mapping $s^c = \text{Epi\_FOXA1}^+ \rightarrow s^t = \text{EMT-Epi1\_CEACAM6}^+$, where FOXA1-positive luminal epithelial cells transition to CEACAM6-expressing EMT-associated states that exhibit enhanced invasive potential. Second, proliferation activation is captured by $s^c = \text{Epi\_FOXA1}^+ \rightarrow s^t = \text{Epi\_CENPF}^+$, representing the transition from quiescent to proliferative epithelial states through CENPF upregulation, a key centromere protein associated with cell cycle progression. Third, lineage conversion is modeled as $s^c = \text{Epi\_FOXA1}^+ \rightarrow s^t = \text{mgEpi\_KRT14}^+$, capturing the luminal-to-basal epithelial transition characterized by KRT14 expression, which is associated with increased stemness and therapeutic resistance.

The immune infiltration axis ($\mathcal{T}_{immune}$) covers transitions modeling the shift from an immune-cold tumor microenvironment to an immune-hot one. T-cell activation is represented by $s^c = \text{tcm\_CD4}^+\text{T} \rightarrow s^t = \text{eff\_CD8}^+\text{T1}$, modeling the functional transition from central memory CD4+ T cells to effector CD8+ T cells, which represents a shift from immunosuppressive to cytotoxic immune responses. Angiogenesis activation follows $s^c = \text{EC\_CAVIN2}^+ \rightarrow s^t = \text{EC\_CLEC14A}^+$, capturing endothelial cell activation from CAVIN2-expressing quiescent states to CLEC14A-positive angiogenic states that facilitate immune cell infiltration and vascular remodeling within the tumor microenvironment.

**Quality Control Criteria.** We impose several biological constraints to ensure that paired transitions are valid: 1) The control and target must belong to the same developmental lineage ($\mathcal{L}(s^c) = \mathcal{L}(s^t)$) to avoid biologically implausible pairings (e.g., an epithelial cell paired with an immune cell). 2) Each cell state must have a minimum number of cells (at least $\theta_{min} = 50$ for both the control state $s^c$ and target state $s^t$) to ensure robust statistical support for the pairing. 3) We preferentially pair cells that reside in similar niches (i.e., $m_i \approx m_j$ in terms of microenvironment), since cellular transitions often occur within the same or adjacent spatial regions.

### D.2 OPTIMAL TRANSPORT-BASED CELL PAIRING

Xenium platform provides paired imaging and gene expression for individual cells in tissue, but we cannot observe the same cell before and after a perturbation. We therefore construct *pseudo* pre-perturbation to post-perturbation examples at the population level. Within each tissue sample, we pair cells from a control state with cells from a target state, enforcing the lineage and spatial constraints above. Importantly, we perform this pairing in the gene expression space (not directly on image features) to avoid any trivial matching based on morphological features and to ensure the pairing reflects underlying molecular state changes that drive morphology.

**Expression Space Preprocessing.** For each defined transition $(s^c \rightarrow s^t)$, we first extract the corresponding subsets of cells $\mathcal{C}^c = i : s_i = s^c$ and $\mathcal{C}^t = j : s_j = s^t$ from the dataset. To address the high dimensionality of gene expression while preserving biological signal, we apply principal component analysis (PCA) to the expression profiles. We center each gene expression vector by the global mean $\bar{g} = \frac{1}{N} \sum_{i=1}^N g_i$ and project it onto the top $d = 50$ principal components. This yields a reduced-dimension representation $\tilde{g}i = \text{PCA}d(g_i - \bar{g})$ for each cell, capturing the majority of expression variance in a compact form that is amenable to efficient computation.

**Sinkhorn-Knopp Algorithm.** We formulate the cell pairing problem as an optimal transport task, leveraging the principle that biologically similar cells should be preferentially paired based on their expression profiles. Given control cells $\{\tilde{g}_i^c\}_{i \in \mathcal{C}^c}$ and target cells $\{\tilde{g}_j^t\}_{j \in \mathcal{C}^t}$, we compute the pairwise cost matrix as

$$C_{ij} = \|\tilde{g}_i^c - \tilde{g}_j^t\|_2, \tag{8}$$

representing the Euclidean distance between expression profiles in the reduced dimensional space.

The optimal transport plan $P^* \in \mathbb{R}^{|\mathcal{C}^c| \times |\mathcal{C}^t|}$ is obtained by solving the entropy-regularized optimal transport problem:

$$P^* = \arg \min_{P \in \Pi(\mu, \nu)} \langle P, C \rangle + \epsilon H(P) \tag{9}$$

where $\Pi(\mu, \nu)$ denotes the set of transport plans between uniform distributions $\mu$ and $\nu$, $H(P) = -\sum_{ij} P_{ij} \log P_{ij}$ is the entropy regularizer promoting smooth transport plans, and $\epsilon = 0.05$ is the regularization parameter that balances transport cost and entropy.

We solve this optimization problem using the Sinkhorn-Knopp algorithm with log-domain updates to ensure numerical stability. The log-sum-exp operation and the transport plan is recovered as:

$$P^* = \exp \left( \left( -C + u^* \mathbf{1}^T + \mathbf{1} v^{*T} \right) / \epsilon \right) \tag{10}$$

From the learned transport plan $P^*$, we derive discrete one-to-one cell pairings for our dataset. For each control cell $i \in \mathcal{C}^c$, we identify its optimal target match by taking the highest probability assignment: $\pi(i) = \arg \max_{j \in \mathcal{C}^t} P_{ij}^*$.

This yields a set of paired indices $(i, \pi(i)) : i \in \mathcal{C}^c$ that approximates the minimum transport cost matching between control and target cells (subject to the constraints encoded in $P^*$), ensuring each control cell is paired with exactly one target cell.

### D.3 DIFFERENTIAL EXPRESSION SIGNATURE COMPUTATION

For each transition type $\tau$ (e.g., each defined axis or process like EMT or T-cell activation), we compute a population-level differential expression signature to characterize the typical gene expression changes associated with that transition. Specifically, let $\mathcal{P}\tau$ be the set of all control-target pairs $(i, j)$ assigned to transition $\tau$ in our paired dataset. We define the signature vector as the average expression change over those pairs:

$$\Delta g_\tau = \frac{1}{|\mathcal{P}_\tau|} \sum_{(i,j) \in \mathcal{P}_\tau} (g_j^t - g_i^c) \tag{11}$$

where $\mathcal{P}_\tau$ represents all pairs of transition type $\tau$. This population-averaged signature provides a robust estimate of the expected gene expression changes during each biological transition, serving as a conditioning signal for the generative model that encodes the molecular mechanisms underlying morphological changes.

### D.4 DATASET CONSTRUCTION PIPELINE

The complete dataset construction process is formalized in Algorithm 1, which integrates biological constraints, optimal transport theory, and quality control measures to generate biologically meaningful perturbation pairs.

### D.5 EXPERIMENTAL DESIGN

**Dataset Characteristics.** Our methodology was applied to the Xenium breast cancer spatial transcriptomics dataset, which provides comprehensive single-cell resolution data with matched morphological information. The dataset contains 165,423 individual cells profiled across 70,611 genes, with 48 distinct cell state annotations derived from expert curation. Each cell is associated with high-resolution H&E histology images that capture morphological features at single-cell resolution, enabling direct correlation between transcriptional states and cellular morphology.

**Generated Perturbation Pairs.** The biologically-informed pairing pipeline successfully generated 1,584 perturbation pairs across two primary biological tasks. Task 1 (Tumor Progression) yielded

---

**Algorithm 1** Biologically-Informed Perturbation Pairing

---

**Require:** Dataset $\mathcal{D}$, transition axes $\mathcal{T}$, parameters $\theta_{\min}$, $\epsilon$, $d$
**Ensure:** Paired dataset $\mathcal{P}$, signatures $\{\Delta g_\tau\}$

1:   $\mathcal{P} \leftarrow \emptyset$
2: **for** each transition $\tau = (s^c \rightarrow s^t) \in \mathcal{T}$ **do**
3:      $\mathcal{C}^c \leftarrow \{\, i \mid s_i = s^c \text{ and } |\{k \mid s_k = s^c\}| \geq \theta_{\min} \,\}$
4:      $\mathcal{C}^t \leftarrow \{\, j \mid s_j = s^t \text{ and } |\{k \mid s_k = s^t\}| \geq \theta_{\min} \,\}$
5:      **if** $|\mathcal{C}^c| = 0$ **or** $|\mathcal{C}^t| = 0$ **then**
6:         **continue**
7:      **end if**
8:      $G^c \leftarrow \text{PCA}_d\big(\{g_i : i \in \mathcal{C}^c\}\big)$                 $\triangleright$ PCA projection
9:      $G^t \leftarrow \text{PCA}_d\big(\{g_j : j \in \mathcal{C}^t\}\big)$
10:    $C \leftarrow \text{compute\_cost\_matrix}(G^c, G^t)$           $\triangleright$ Optimal transport
11:    $P^* \leftarrow \text{sinkhorn\_knopp}(C, \epsilon)$
12:    $\pi \leftarrow \text{hard\_assignment}(P^*)$
13:    **for** $i \in \mathcal{C}^c$ **do**                       $\triangleright$ Generate pairs
14:       $j \leftarrow \pi(i)$
15:       $\mathcal{P} \leftarrow \mathcal{P} \cup \{(x_i, x_j, g_i, g_j, \tau)\}$
16:    **end for**
17:    $\Delta g_\tau \leftarrow \text{mean}\big(\{\, g_j - g_i : (i,j) \in \text{pairs of type } \tau \,\}\big)$     $\triangleright$ Compute signature
18: **end for**
19: **return** $\mathcal{P}, \{\Delta g_\tau\}$

---

798 pairs distributed across three biological processes: EMT transition (266 pairs), proliferation activation (266 pairs), and lineage conversion (266 pairs). Task 2 (Immune Infiltration) produced 786 pairs spanning two processes: T-cell activation (400 pairs) and angiogenesis activation (386 pairs). This distribution reflects both the natural abundance of different cell states in the breast cancer tissue and our balanced sampling strategy to ensure sufficient statistical power for each transition type while maintaining biological authenticity. For example, a pairing result may look like

{x_ctrl_id, x_tgt_id, state_A, state_B, cell_type, niche_ctrl, niche_tgt, transition_tag, task_name, patient_id, slide_id, spatial_distance_um, match_score}:

{100119,131051, Epi_FOXA1+, EMT-Epi1_CEACAM6+, Epithelial, Epi-Immune, EMT-Immune, EMT_transition, tumor_progression, P001, S07, 38, 0.87}

### D.6 MORE RELATED WORK

**Single Cell Models.** Foundation models for single-cell (non-spatial) transcriptomics have rapidly advanced, leveraging large-scale pretraining to support diverse downstream tasks such as cell type annotation, gene network inference, and perturbation prediction Cui et al. (2023); Yang et al. (2022). Notable models include scGPT Cui et al. (2023), scBERT Yang et al. (2022), scPRINT Kalfon et al. (2025), scMulan Bian et al. (2024), scFoundation Hao et al. (2024a), scInterpreter Li et al. (2024a), scHyena Oh et al. (2023), GET Fu et al. (2025a), SCimilarity Heimberg et al. (2024), and xTrimoGene Gong et al. (2023). These models are pretrained on repositories encompassing tens to hundreds of millions of cells, allowing them to capture complex transcriptional grammars, gene regulatory networks, and cellular heterogeneity across diverse biological contexts Hao et al. (2024a); Fu et al. (2025a). However, they focus on transcriptomic data, lacking integration with spatial or imaging modalities, which are crucial for understanding cellular context within tissues.

## E MORE EXPERIMENTS AND ABLATION

**Cross-modal prediction.** We train a MLP decoder to reconstruct held-out gene expression $g_i$ from cell embeddings $\mathbf{C}_i$ and, conversely, to predict cell embeddings from gene expression inputs. Reconstruction quality is measured via Pearson or Spearman correlation between predicted $\hat{\mathbf{g}}_i$ (or $\hat{\mathbf{C}}_i$) and ground truth.

Table 11: Performance on cross-modal prediction and generation tasks.

| Task | Cross modal Pred. | | Cross modal Gen. | |
|---|---|---|---|---|
| | Pearson ↑ | Spearman ↑ | PSNR ↑ | SSIM ↑ |
| SPATIA | 0.43 | 0.41 | 24.80 | 0.65 |

**Niche level effectiveness.** We additionally trained a cell-only variant of SPATIA that preserves the same cell-level image encoder, gene encoder, and conditional flow module, but removes niche/tissue embeddings and multi-scale spatial fusion. This ablation isolates the effect of spatial context while keeping architectural capacity comparable.

Table 12: Effect of removing niche-level context on conditional generation

| Method | Image fidelity | | Morphology correctness | |
|---|---|---|---|---|
| | FID ↓ | KID ↓ | Wass. Corr. ↑ | KS ↑ |
| SPATIA w/o niche level | 64.3 | 2.44 | 0.87 | 0.56 |
| SPATIA | 59.2 | 2.04 | 0.92 | 0.62 |

**Scaling Evaluation.** To analyze whether SPATIA benefits from larger backbone capacity, we evaluated multiple Vision Transformer (ViT) variants while keeping the gene-expression encoder fixed (as scPRINT is currently among the few pretrained models for scRNA-seq). We compared a Base and a Large encoder variant.

Table 13: Scaling behavior of ViT-based image encoders

| Vision Encoder | Params (M) | Loss | | Clustering | |
|---|---|---|---|---|---|
| | | Train ↓ | Val ↓ | ARI ↑ | NMI ↑ |
| SPATIA-ViT-Base | 86M | 0.4620 | 0.4518 | 0.870 | 0.831 |
| SPATIA-ViT-Large | 307M | 0.4885 | 0.4637 | 0.842 | 0.805 |

Interestingly, the larger ViT model performs worse than the Base variant. We attribute this decline to two factors: (1) natural-image priors inherited by larger ViTs transfer poorly to fluorescence-based morphology data, and (2) despite dataset scale, spatial single-cell assay variability remains limited relative to natural-image corpora, leading to overfitting of fine-grained noise rather than meaningful structure.

**Importance of Pretraining for Image Encoder Initialization.** To quantify the effect of pretrained initialization, we trained SPATIA with a randomly initialized vision encoder and compared it against a version using pretrained weights on morphology patches.

Table 14: Effect of pretrained weights on SPATIA performance

| Vision Encoder Init. | Loss | | Clustering | |
|---|---|---|---|---|
| | Train ↓ | Val ↓ | ARI ↑ | NMI ↑ |
| SPATIA (from scratch) | 0.4838 | 0.4625 | 0.813 | 0.774 |
| SPATIA (from pretrained) | 0.4620 | 0.4518 | 0.870 | 0.831 |

Pretrained initialization substantially improves optimization stability, convergence, and downstream clustering quality. This suggests that incorporating priors from morphology-aware pretraining is crucial for reliable representation learning under limited morphological variation.

