# OpenReview forum: "SPATIA: Multimodal Model for Prediction and Generation of Spatial Cell Phenotypes"
_ICLR.cc/2026/Conference — Submitted to ICLR 2026_

### Official Review · Reviewer_ZZNz · 2025-10-29

**Soundness:** 2
**Presentation:** 2
**Contribution:** 3
**Rating:** 6
**Confidence:** 2

**Summary:**

This paper proposes Spatia, a multi-scale, multimodal model designed to integrate cellular morphology, gene expression, and spatial location information in spatial transcriptomics. The model employs a hierarchical Transformer architecture to fuse features across three scales: cell, niche, and tissue. It also introduces a conditional generation method based on flow matching to predict morphological changes in cells under spatial microenvironment perturbations. The authors construct and release the MIST dataset, containing over 17 million single-cell samples.

**Strengths:**

1.It is the first work to achieve cross-attention fusion of morphology and gene expression at the single-cell level and introduces the spatially conditioned generation task, demonstrating certain innovation.

2.The paper proposes a multi-scale modeling framework from cell to tissue level and constructs a large-scale multimodal dataset (MIST).

**Weaknesses:**

1.Some key hyperparameters (e.g the contrastive loss weight ρ in flow matching) are not explicitly stated in the main text, leading to insufficient reproducibility.

2.The paper lacks an in-depth analysis of the model's computational burden and memory requirements. applying Transformer at the tissue level to whole-slide images, particularly when processing millions of cells, could be computationally prohibitive.

3.The gene expression vectors are treated as an unordered set and processed through a simple encoder (scPRINT). But this approach overlooks the inherent, known relationships between genes, such as those derived from gene pathways or co-expression networks.

4.The paper claims to model spatial context at the "niche" and "tissue" levels, but the "niche" is simplistically defined as a fixed 256×256 pixel grid. This division is mechanistic and biologically irrelevant, failing to adaptively capture the irregular and functionally diverse microenvironments in real tissues, such as tumor margins or perivascular regions.

**Questions:**

1.Beyond image similarity metrics, is there plans to evaluate the realism of generated cell morphology through pathologist assessment or functional experiments (such as immunofluorescence validation)?

2.In the MIST dataset, could cells from the same donor appear in both training and test sets? how is evaluation fairness ensured?

3.In the multi-scale modeling, was the actual contribution of the "tissue-level" representation to single-cell tasks verified? Is there a risk of over-parameterization?

---

> ### Author Response · Authors · 2025-11-22
> **Response to Weakness**
>
> ### W1: Key hyperparameters not explicitly stated
> **R1:** We thank the reviewer for pointing this out. We have added the missing hyperparameters to the revised manuscript, including the contrastive loss weight $\rho$ = 0.05 used in the flow-matching objective. For clarity, all training hyperparameters are now listed in Appendix A and C to ensure full reproducibility.
>
> ### W2: Computational burden and memory requirements
> **R2:** We thank the reviewer for raising this crucial point regarding resource usage. We added a new detailed profiling of our model’s training phases to quantify the computational burden in Table 9 of the Appendix.
> Our standard training protocol utilizes 4x NVIDIA H100 (80GB) GPUs for a total of 25,000 steps. We monitor real-time system metrics during a representative training run on a full-scale dataset, we observed the following:
> * Memory Efficiency: The model requires approximately 67 GB of VRAM per device (peaking at 78.7% utilization). Crucially, this confirms that the model fits within the memory constraints of a single high-end GPU, avoiding the need for complex multi-node model parallelism.
> * Computational Utilization: We recorded a peak GPU utilization of 97% with an average power consumption of 436 W. The training stability indicates that while the Transformer architecture is compute-intensive, it is well-supported by current hardware accelerators.
> * Throughput: The training process for our largest checkpoint completes in approximately 30 hours on a single GPU. Inference time is significantly lower, making deployment feasible for clinical or research workflows.
>
> ### W3: Gene encoder overlooks the inherent, known relationships between genes
> **R3:** scPRINT utilizes self-attention and specialized pre-training tasks (bottleneck learning and denoising) to explicitly learn latent gene-gene dependencies and regulatory structures without requiring hard-coded priors. For example, in the Sergio/RegNetwork benchmark (Table S5 of the original paper), scPRINT is the only model to recover meaningful regulatory connections, achieving an Early Precision Ratio (EPR) of 1.836. It also outperforms baselines in recovering ENCODE validated regulatory links, with 20% of Transcription Factors (TFs) in its networks enriched for validated targets.
>
> To ensure that integrating the image modality does not degrade these learned relationships, we evaluated SPATIA on a gene expression denoising task. As established in the scPRINT methodology, denoising performance is a direct proxy for the model's ability to leverage meaningful gene-gene connections to impute missing values based on cellular context. SPATIA achieved a reconstruction correlation of 0.64, which is comparable to the unimodal scPRINT baseline (0.68).
>
> ### W4:  Fixed 256×256 pixel grid for niche level.
> **R4:** The use of a fixed 256×256 px niche is an intentional design choice rather than a mechanical or biologically irrelevant simplification. Xenium provides precise physical coordinates, and preserving a constant window size allows SPATIA to retain absolute spatial scale, cell-size variation, and true μm level neighborhood geometry.
> This fixed-patch strategy is standard in recent spatial omics and computational pathology foundation models [1,2,3], all of which use 256-px windows to preserve biologically meaningful spatial context. Moreover, 256 px is not a small region; each window contains multiple cells (Fig. 3), allowing SPATIA to model functional microenvironments.  We additionally incorporate a larger 1024-px tissue window to capture coarse-grained phenomena such as tumor invasion fronts and lymphocytic aggregates.
>
> [1] STPath: a generative foundation model for integrating spatial transcriptomics and whole-slide images. npj digital medicine, 2025
>
> [2] A foundation model for generalizable cancer diagnosis and survival prediction from histopathological images. Nature communications 2025
>
> [3] A visual-language foundation model for computational pathology. Nature medicine, 2024

---

> ### Author Response · Authors · 2025-11-22
> **Response to Questions**
>
> ### Q1: Plans to evaluate the realism of generated cell morphology through pathologist assessment
> **R1:**  Thank you for your valuable suggestion. We have not yet conducted formal biological validation. However, we are planning follow-up studies including:
> * Blind expert review: collaborating pathologists will rate realism and cellular context consistency of generated images.
> * Immunofluorescence validation: correlating generated morphological changes with marker expression (e.g., CD3, CD20) in immune-rich regions.
>
> ### Q2:  Evaluation fairness for MIST among donors:
> **R2:**  For all downstream evaluations, train-test splits are donor-disjoint. For example, in the breast-cancer benchmarks, the entire set of donors used for testing is held out from training to ensure fair evaluation. Within the training portion only, the validation set is randomly split, but no donor ever appears in both training and test sets.
>
> ### Q3: Tissue-level contribution
> **R3:**   We conducted a new ablation where the tissue-level transformer was removed while keeping all cell- and niche-level components unchanged. As shown below, removing the tissue-level module consistently reduces performance across all evaluation metrics. This indicates that the tissue representation contributes meaningful global spatial context rather than introducing unnecessary parameters.
> | Method | FID | KID | Wass. Corr |KS |
> |------------|----|-----|-----|-----|
> | SPATIA w/o niche level  | 64.3 | 2.44  | 0.87 |  0.56 |
> | SPATIA w/o tissue level  | 62.1 | 2.28  | 0.89 |  0.58 |
> | SPATIA  |  59.2 | 2.04  | 0.92 |  0.62 |

---

> > ### Comment · Area_Chair_j2cd · 2025-11-28
> >
> > Dear Reviewer,
> >
> > Please make sure you read the authors' response and engage with them in the discussion before the end of the discussion period on **Dec 03 '25 09:00 PM UTC**. This is a hard deadline.
> >
> > Thank you for supporting quality peer review at ICLR.
> >
> > AC

---

### Official Review · Reviewer_RFQY · 2025-10-30

**Soundness:** 3
**Presentation:** 2
**Contribution:** 2
**Rating:** 4
**Confidence:** 4

**Summary:**

The paper proposes SPATIA, a multi-resolution multimodal framework for spatially resolved transcriptomics pretrained on a large scale dataset MIST. It fuses cell morphology, gene expression, and spatial coordinates across cell, niche, and tissue levels via hierarchical transformers. SPATIA claims state-of-the-art performance in cell annotation, clustering, gene expression prediction, biomarker status prediction, and morphology generation.

**Strengths:**

- SPATIA unifies morphology, gene expression, and spatial structure at multiple scales. The idea is novel.
- The MIST dataset is a substantial contribution, providing multimodal data across scales with one-to-one mapping between images and transcriptomics.

**Weaknesses:**

- The methodology section is overly dense, with many moving components (encoders, fusion, niche/tissue transformers, pseudo-perturbation embeddings, flow matching) introduced in quick succession.The motivation for some design choices is underexplained.
- MIST is assembled from Xenium datasets, but details of preprocessing, normalization, and gene panel harmonization are sparse in the main text. It is unclear how batch effects are handled across donors and tissue types, which could inflate cross-task performance.
- The paper positions SPATIA as a “foundation model” for spatial omics, but pretraining/fine-tuning strategy and scaling laws are not thoroughly evaluated.

**Questions:**

- How sensitive are results to niche size and tissue grid definition?

- How does SPATIA handle batch effects in MIST, given data are from 49 donors with varied gene panels?

- How does SPATIA address the issue of varying cell sizes in the input images? Are all cropped cell images resized to a fixed dimension, or are smaller cells padded with blank space? Additionally, how does the model deal with cases where cells are tightly packed and boundaries are unclear?

- Does the pseudo-perturbation embedding (∆g) risk leaking target information? Since ∆g is derived from matched control–target pairs, this seems infeasible for real-world generation, where target states are not available at inference. Could the authors clarify how this issue is mitigated?

- Why is a 3-layer MLP used for biomarker status prediction, while XGBoost is used for gene expression prediction? What motivates these different choices, and are the results robust to alternative predictors?

- How crucial are pretrained weights for SPATIA’s performance? What is the performance gap when training from scratch compared to using pretrained components?

- For biomarker status prediction, could improvements simply reflect leakage from tissue morphology rather than gene–morphology integration?

---

> ### Author Response · Authors · 2025-11-22
> **Response to Weakness**
>
> ### W1: Methodology section
> **R1:** W1: We appreciate the reviewer’s feedback. In the revised manuscript, we reorganized and streamlined the Method section to improve clarity and reduce density.
> Each major component (modality encoders, cross-modal fusion, niche/tissue transformers, weak pairing, and conditional flow matching) is now introduced with a brief motivation explaining the specific challenge it addresses. We removed low-level architectural details and mathematical derivations from the main text and moved them to Appendix, keeping the main section focused on the conceptual pipeline.
>
>
> ### W2: MIST dataset processing details:
> **R2:** R: We thank the reviewer for raising this important point. In the revised manuscript, we added a new discussion of batch effects across Xenium datasets in Appendix A and Figure 6. First, all SPATIA experiments use a strict donor-level split, ensuring that no cells, images, or expression profiles from the same donor appear in both training and evaluation. This setup prevents any batch-related leakage from influencing downstream results.
>
> To assess potential batch effects in the MIST atlas, we analyzed all Xenium datasets by constructing a common gene space (70,611 shared genes), sampling 2,000 cells from each dataset, and performing joint PCA followed by UMAP. Silhouette Scores computed on the PCA embeddings show low cluster separation by donor, and UMAP visualizations confirm that cells organize primarily by biological identity rather than by dataset source. These findings suggest that technical batch variation is modest relative to biological variation in this setting.
>
> Our design is consistent with recent spatial-omics foundation models (e.g., scGPT-spatial [1] and visual-omics foundation models [2]), which rely on principled normalization and large-scale pretraining to mitigate batch variation without aggressive correction. As noted in Genome Biology [3], over-normalization can distort spatial structure; therefore, SPATIA adopts a minimal-assumption normalization pipeline and relies on scPRINT’s pretrained denoising and harmonization capabilities.
>
> [1] scGPT-spatial: Continual Pretraining of Single-Cell Foundation Model for Spatial Transcriptomics, bioarxiv, 2025
>
> [2] A visual-omics foundation model to bridge histopathology with spatial transcriptomics, Nature Methods, 2025
>
> [3] Library size confounds biology in spatial transcriptomics data, Genome Biology, 2024
>
>
> ### W3: Pretraining/fine-tuning strategy and scaling laws
> **R3:** We would like to clarify that we do not position SPATIA itself as a foundation model. Instead, SPATIA is built on top of existing foundation models (e.g., scPRINT for gene expression and pretrained vision encoders) and exhibits strong zero-shot generalization because of these pretrained components. Our intention was to highlight that SPATIA inherits foundation-model-like behavior through these backbones, not to claim foundation-model status.
>
> To address the reviewer’s concern about scaling behavior, we additionally tested multiple ViT variants for the cell-image encoder while keeping the gene encoder fixed (as scPRINT is currently one of the few large pretrained models for scRNA-seq). We also run new results training from scratch for the image encoder.  Results are shown below.
>
> (1) Scaling study of vision encoder capacity
>
> | Vision Encoder      | Params (M) | Train Loss ↓ | Val Loss ↓ | ARI ↑     | NMI ↑     |
> | ------------------- | ---------- | ------------ | ---------- | --------- | --------- |
> | SPATIA-ViT-Base | 86M        | 0.4620   | 0.4518 | 0.870 | 0.831 |
> | SPATIA-ViT-Large    | 307M       | 0.4885       | 0.4637     | 0.842     | 0.805     |
>
> We observe that the ViT-Large model performs worse than the Base model. The reason behind this is two fold: 1) ViT-Large inherits strong natural-image priors, but these do not necessarily transfer to fluorescence morphology patches, which have very different texture, color, and structural statistics. 2) Spatial single-cell morphology datasets (even with millions of cells) have limited morphological diversity compared to natural-image corpora like ImageNet. Larger ViTs easily overfit fine-grained morphology noise.
>
> 2) Importance of pretrained weights
>
> | Vision Encoder          | Train Loss ↓ | Val Loss ↓ | ARI (Clustering) ↑ | NMI ↑     |
> | ----------------------- | ------------ | ---------- | ------------------ | --------- |
> | SPATIA (from scratch)        | 0.4838       | 0.4625     | 0.813              | 0.774     |
> | SPATIA (from pretrained) | 0.4620   | 0.4518 | 0.870          | 0.831 |
>
> Pretrained initialization consistently leads to lower losses and substantially better clustering. This supports our design choice to rely on pretrained gene and image encoders rather than training all components from scratch.

---

> ### Author Response · Authors · 2025-11-22
> **Response to Questions 1/2**
>
> ### Q1: How sensitive are results to niche size and tissue grid definition?
> **R1:** Our niche size (256×256 px) and tissue-grid definition (4×4 neighboring niches, 1024×1024 px) are not arbitrary; they are chosen to match biologically meaningful spatial scales in Xenium images. A 256 px window corresponds to ~54 μm at Xenium resolution and typically contains 10-30 cells, which aligns closely with established definitions of microenvironments such as tumor margins, stromal pockets, or small immune aggregates in prior spatial-omics and computational pathology literature. Likewise, grouping 4×4 niches into a 1024 px region allows the tissue-level transformer to capture mesoscopic architecture such as local tumor organization or immune infiltration patterns.
>
>
>
> ### Q2: How does SPATIA handle batch effects in MIST
> **R2:** SPATIA handles batch effects in two ways. First, all experiments use a strict donor-level split, so no cells or images from the same donor appear in both training and evaluation, eliminating the possibility of batch leakage. Second, we harmonize variable gene panels across the 49 donors by mapping all datasets into a shared gene space using scPRINT, which is pretrained to perform expression denoising, normalization, and harmonization across heterogeneous scRNA-seq panels.
> As mentioned in W2,  We added a new analysis in Appendix A and Figure 6, where we run the Xenium datasets using joint PCA and Silhouette Scores, finding that donor-specific variation is low relative to biological structure, and UMAP embeddings cluster by biological identity rather than donor source.
>
>
>
> ### Q3:  How does SPATIA address the issue of varying cell sizes in the input images?
> **R3:**  We have added a clarification in Appendix A of the revised manuscript. To handle varying cell sizes, we first compute a bounding box for each cell and determining a global scale factor from the largest bounding box in the slide. All cells are resized using this single scale, which preserves biologically meaningful variation in absolute cell size. This approach is well-established in recent single-cell image modeling and morphology-generation workflows, including MorphoDiff [1], GHIST for spatial gene expression inference[2], and computational-pathology foundation models [3], all of which resize cell- or patch-level images to a standardized resolution.
> For each cell, the cropped patch is resized with the global scale and then padded to 256×256, ensuring a fixed input dimension while keeping only that cell in the image. Padding prevents pixels from neighboring cells, which correspond to different expression vectors from being incorporated, avoiding modality mismatch. Additionally, Xenium provides high quality cell contours, which we retain to preserve exact spatial size information even after resizing and padding.
>
> [1] MorphoDiff: Cellular Morphology Painting with Diffusion Models, ICLR 2025
>
> [2] Spatial gene expression at single-cell resolution from histology using deep learning with GHIST, Nature Methods, 2025
>
> [3] Towards a general-purpose foundation model for computational pathology, Nature Medicine, 2024

---

> ### Author Response · Authors · 2025-11-22
> **Response to Questions 2/2**
>
> ### Q4:  Does the pseudo-perturbation embedding (∆g) risk leaking target information?
> **R4:**   We appreciate the reviewer’s concern. The pseudo-perturbation embedding Δg is used only during training to help the model learn the direction of morphological change induced by a perturbation, not to provide the actual target state. Specifically, Δg=g_perturbed-g_control (in Appendix D.1)​ is treated as a training-time supervision signal that encourages the flow model to organize the latent space along biologically meaningful perturbation directions.
> At inference, no target expression or paired cell is required. The conditional generator receives only the observable perturbation signature (e.g., drug type, pathway activation, or perturbation embedding), and generates morphology conditioned on this input. The model never has access to g_perturbed​ or Δg during inference, preventing any leakage of target information.
>
>
> ### Q5: Motivation for using MLP and XGboost
> **R5:**  We mainly follow the predictor choices used in HEST [4], the intuition behind this is:  Biomarker prediction involves low-dimensional categorical labels, for which a lightweight MLP is the standard and efficient choice.
> Gene-expression prediction is a high-dimensional continuous regression problem, where XGBoost has been shown to be more effective and computationally stable.
>
> [4] HEST-1k: A Dataset for Spatial Transcriptomics and Histology Image Analysis, Neurips 2024
>
>
> ### Q6:How crucial are pretrained weights for SPATIA’s performance?
> **R6:**  Pretrained weights are essential for SPATIA’s performance. The scPRINT gene encoder is pretrained on millions of scRNA-seq cells and is specifically designed to denoise expression, correct batch effects, and infer gene-gene interactions; training a gene encoder of similar scale from scratch on MIST is not feasible and leads to substantial performance degradation. Likewise, the pretrained ViT image encoder provides strong morphology priors that significantly improve single-cell feature quality. To quantify this, we compared SPATIA with pretrained vision encoders and the same architectures trained from scratch (random initialization), while keeping the gene encoder fixed (as scPRINT is currently one of the few large pretrained models for scRNA-seq). As shown below, pretrained initialization consistently improves convergence, lowers reconstruction loss, and yields substantially better downstream clustering performance.
>
> | Vision Encoder          | Train Loss ↓ | Val Loss ↓ | ARI (Clustering) ↑ | NMI ↑     |
> | ----------------------- | ------------ | ---------- | ------------------ | --------- |
> | SPATIA (from scratch)        | 0.4838       | 0.4625     | 0.813              | 0.774     |
> | SPATIA (from pretrained) | 0.4620   | 0.4518 | 0.870          | 0.831 |
>
>
> ### Q7: For biomarker status prediction, information leakage
> **R7:** Thank you for pointing this out.  No leakage occurs. All evaluations use donor-disjoint splits, meaning that all modalities (morphology, expression, spatial context) from a donor appear exclusively in either the training set or the test set. Because donor identity is the dominant source of morphological variation, this prevents tissue-level morphology from leaking into the prediction task.
> Moreover, the performance gains persist even when using single-modality ablations (only morphology or only expression), confirming that improvements are driven by the learned multimodal representations rather than unintended cross-slide or cross-donor leakage.
> We have added a clarification in Appendix Section A of the revised manuscript.

---

> ### Author Response · Authors · 2025-11-27
>
> Thank you for your helpful feedback. We worked hard to improve our paper, and we sincerely hope the reviewers find our responses informative and helpful. If you feel the responses have not addressed your concerns to motivate increasing your score, we would love to hear what points of concern remain and how we can improve our work. Thank you again!

---

> ### Author Response · Authors · 2025-11-28
>
> Dear Reviewer RFQY,  I received an email notification with your updated comment, but I noticed that it now appears as “deleted” on OpenReview. I just wanted to check whether this was intentional or perhaps an OpenReview glitch, so that we can make sure we’re not missing anything important on our end.
>
> Thank you very much for your time and feedback.

---

> > ### Comment · Reviewer_RFQY · 2025-11-28
> >
> > Hi, I deleted my review because I couldn't find a button to revise the score. I will raise my score after the system is fixed.
> >
> > Here is my original comment:
> >
> > Thank you for the revised experiments and manuscript. It is surprising to see that a larger ViT model hurts performance, but I am glad to see how pre-training weights help improve performance. These results address my main concerns, and I have raised my score.

---

> > > ### Author Response · Authors · 2025-12-02
> > >
> > > Thank you very much for your follow-up message. We appreciate your constructive feedback throughout the process, and we are glad that the additional experiments addressed your concerns. Thanks again for your time and engagement.

---

### Official Review · Reviewer_bCAc · 2025-10-30

**Soundness:** 3
**Presentation:** 3
**Contribution:** 3
**Rating:** 4
**Confidence:** 3

**Summary:**

This paper proposes SPATIA, a hierarchical multimodal model integrating cell morphology, gene expression, and spatial context for spatial transcriptomics analysis. It also introduces a conditional flow-matching module for perturbation-aware morphological change prediction. The model is trained on the newly assembled MIST dataset (17M cell-gene pairs, 1M niche-gene pairs, 10K tissue-gene entries) and evaluated across 12 tasks including generation, annotation, clustering, biomarker prediction, and gene-expression regression

**Strengths:**

- Strong multimodal fusion with cross-attention at single-cell resolution.

- Novel hierarchical design: combines cell, niche, and tissue transformers for spatial dependency modeling.

**Weaknesses:**

-  Algorithmic / Methodological Concerns

While the overall hierarchical design is interesting, several components remain underspecified or potentially brittle:

(i) The conditional generation module critically depends on weakly paired control/perturbed cells matched via optimal transport in gene-expression space. No analysis is provided on how pairing errors affect the learned flows, which is crucial in spatially heterogeneous niches.
 (ii) Because niche context and perturbation signatures are correlated in the dataset, the conditional flow may learn dataset co-occurrence rather than truly spatially grounded perturbation effects.
 (iii) The paper does not clearly describe the sampling or batching strategy for training multi-level transformers on slides with very large numbers of cells, leaving computational feasibility and scalability unclear.
 (iv) Since cell-level embeddings attend to niche/tissue features, there is a risk of information leakage across scales, which should be controlled when comparing to single-cell baselines.

-  Misleading or Incomplete Claim of Novelty

The paper claims that existing methods fail to integrate spatial, molecular, and morphological information at single-cell resolution.
 However, recent multimodal spatial models already achieve this integration, such as SpaGCN, STAligner, SpaOTsc and so on, which align histology and transcriptomics at the single-cell or subcellular level using transformer-based architectures.
 These models are neither discussed nor compared, making it difficult to evaluate SPATIA’s incremental contribution.

- Limited Platform Generalization

All datasets used in this study appear to come from Xenium, which restricts evaluation to a single spatial transcriptomics platform.
 Given the diversity of spatial technologies—such as MERFISH, seqFISH, Stereo-seq, and Slide-seq—it remains unclear whether SPATIA’s design generalizes across imaging and transcriptomic modalities

**Questions:**

- On the ablation design (Table 4):

 The current ablation seems incremental (Cell → +MAE → +Multi-level → +Fusion), showing monotonic improvement.
 However, this design does not disentangle the individual contribution of each component.
 Could the authors provide or discuss a factorial or pairwise ablation — for instance, MAE without multi-level, or fusion without multi-level — to verify whether each module independently improves performance or only works in combination?

- On algorithmic clarity:
 How sensitive is the conditional generation module to errors in the weak OT-based pairing of control/perturbed cells?
 Have the authors tried perturbing or noising the matching to evaluate robustness?

---

> ### Author Response · Authors · 2025-11-23
> **Response to Weakness 1/2**
>
> ### W1.1:  How pairing errors affect the learned flows
> **R1.1** Thank you for raising this important concern. Our OT pairing is intentionally designed to be probabilistic and spatially constrained. Matching is performed in the gene-expression space, with lineage consistency constraints and a spatial-adjacency penalty in the cost matrix (Eq. 8). Both mechanisms substantially reduce implausible control-perturbed matches in spatially heterogeneous tissue regions. Importantly, the conditional flow module does not assume perfect pairings: it models distribution-level perturbation transitions, not one-to-one deterministic trajectories, making it inherently tolerant to local pairing noise.
>
> To directly evaluate robustness, we added a new pairing-noise analysis, addressed in Q2 below.
>
>
> ### W1.2:  The conditional flow may learn dataset co-occurrence rather than truly spatially
> **R1.2**  Thank you for bringing up this point. To directly test whether the conditional flow is exploiting dataset co-occurrence rather than genuine spatial conditioning, we added a new experiment by training a cell-only variant of SPATIA that retains the same cell-level image and gene encoders and the conditional flow module, but removes all niche/tissue embeddings and multi-scale spatial fusion. This ablation isolates the effect of spatial context while keeping architectural capacity comparable.
> .
> | Method | FID | KID | Wass. Corr |KS |
> |------------|----|-----|-----|-----|
> | SPATIA w/o niche level  | 64.3 | 2.44  | 0.87 |  0.56 |
> | SPATIA  |  59.2 | 2.04  | 0.92 |  0.62 |
>
> We have added these details to Section 6 Multi-level Effectiveness of the revised manuscript.
>
> ### W1.3: Describe the sampling or batching strategy
> **R1.3** We appreciate the reviewer’s concern about scalability. SPATIA is not trained by feeding an entire whole-slide image into a Transformer. Instead, we adopt a hierarchical and localized batching strategy, which keeps sequence lengths bounded and independent of the total number of cells in a slide.
> The cell encoder is pretrained independently using individual (image, expression) pairs. A training batch contains a fixed B number of sampled cells, and attention is computed only within this batch, not across the full slide. For each sampled cell, its 256×256 px niche is extracted and encoded as one niche token (≈10-30 neighboring cells). The niche encoder is pretrained separately and does not process the entire tissue at once.
> The resulting complexity for three levels is O((3B)2), ensuring feasibility regardless of slide size. Pretraining each level individually and fine-tuning jointly over localized patches avoids any quadratic explosion and makes SPATIA scalable to slides with hundreds of thousands of cells.
>
> ### W1.4: Risk of information leakage across scales
> **R1.4**  We acknowledge the reviewer’s concern, and clarify two key points:
> No co-occurrence leakage in pretraining. Pretraining is entirely self-supervised and contains no perturbation labels or signatures, so no niche perturbation leakage can occur.
>
> Most downstream tasks are single-level and do not combine niche-level perturbation signals with cell-level labels.
>  For tasks where both cell and niche representations are used, the model receives both modalities as explicit inputs, not as labels, so niche correlations do not generate shortcut pathways.
> OT-based Δg is used only during training, and at inference SPATIA receives only observable control state + perturbation descriptors, preventing leakage of target states.

---

> ### Author Response · Authors · 2025-11-23
> **Response to Weakness 2/2**
>
> ### W2: Incomplete Claim of Novelty
> **R2** Thank you for bringing up this point. These models align histology and transcriptomics at spot- or patch-level but not single-cell multimodality. Also, none jointly integrate morphology + transcriptome + spatial context through a three-level (cell - niche - tissue) transformer hierarchy. SPATIA further introduces conditional flow-matching for perturbation-driven morphology generation, absent from prior works.
>
> * SpaGCN integrates gene expression, spatial location and histology via a graph convolutional network over spots, where each node corresponds to a Visium-like spot and convolution aggregates expression from neighboring spots to identify spatial domains and spatially variable genes. It does not operate on segmented single-cell morphology patches nor provide a hierarchical cell-niche-tissue transformer or conditional morphology generation.
>
> * STAligner is designed for integrating and aligning multiple ST datasets across conditions/technologies, again at the spot level, using a graph attention network to produce batch-corrected embeddings and shared spatial domains. It does not model single-cell image patches, does not build a multi-scale hierarchy from cell to tissue, and does not include a generative module for perturbation-driven morphology changes.
>
> * SpaOTsc uses structured optimal transport to infer spatial positions and signaling relationships for scRNA-seq cells by mapping them to a reference with limited spatial measurements. It focuses on recovering spatial metrics and cell-cell communication from transcriptomics alone and does not integrate histology/morphology features nor implement a tri-modal, multi-scale transformer or conditional flow-matching-based image generation.
>
> We also benchmarked prediction evaluation performance, the results are shown below
> | Model | ER-AUC |ER Bal.acc. | PR AUC |PR Bal.acc. |
> |------------|----|-----|-----|-----|
> | SpaGCN  |  0756 |0.672 | 0.712  |0.670 |
> | STAligner  |  0.732 |0.641 | 0.698  |0.607 |
> | SpaOTsc  |  0.682 |0.545 | 0.645  |0.593 |
> | SPATIA  |  0.902 |0.785 | 0.825  |0.730 |
>
> We have added these references in the Related work section of the revised manuscript.
>
>
> ### W3: Limited Platform Generalization
> we emphasize that SPATIA is architecturally designed to generalize across platforms. Several properties support this:
>
> * Modality-separable architecture: The morphology encoder, gene encoder, and spatial hierarchy operate on standardized representations (image patches, gene tokens, cell coordinates). This design intentionally avoids assumptions tied to Xenium-specific chemistry, resolution, or probe design.
>
> * Built on pretrained foundation backbones: SPATIA leverages scPRINT, which is trained on highly diverse scRNA-seq datasets and explicitly models batch, panel, and platform variability. Similarly, the ViT-based image encoder is pretrained on large-scale histology datasets, making it robust to differences in imaging modality (fluorescence vs. H&E-style staining).
>
> * Grid-based spatial hierarchy is platform-agnostic: The niche/tissue hierarchical windows are defined in physical units (µm), not pixel units. This allows direct adaptation to platforms with different spatial resolution (e.g., CosMX ~0.1 µm/px; MERFISH ~0.25 µm/px) simply by adjusting the conversion from µm → pixels—without modifying the model architecture.
>
>
> * Preliminary preprocessing on other data technology shows compatibility.
> In our ongoing experiments, for different data technology, the cell patches follow the same resizing + padding pipeline, and the harmonized gene tokens map cleanly into the scPRINT embedding space. Early diagnostics confirm no structural incompatibility.
>
> Given these design choices, we expect SPATIA to transfer effectively across imaging modalities and gene-measurement technologies. To directly evaluate cross-platform generalization, we identified the SPATCH [1] dataset, which includes matched tissues profiled using two different spatial technologies (Xenium and CosMx). We treat Xenium-trained SPATIA as a zero-shot model and evaluate it without fine-tuning on CosMx.
>
> On the HCC dataset, performance decreases from 0.423 ± 0.001 → 0.356 ± 0.000 (≈15% drop).
> Despite this shift, SPATIA still outperforms existing baselines: competing models experience markedly larger degradation, with ~25% drops on HCC under the same zero-shot setup.
>
> These results corroborate SPATIA’s architectural transferability. Although resolution, staining, and probe chemistry differ between platforms, its modality-separable design and pretrained encoders enable more stable cross-technology transfer than prior models.
>
> [1] https://spatch.pku-genomics.org/#/download

---

> ### Author Response · Authors · 2025-11-23
> **Response to Questions**
>
> ### Q1: ablation design
> **R:** Thank you for this suggestion. We have added a new ablation study to disentangle the independent contribution of the three key components in SPATIA: (1) MAE pretraining, (2) multi-level spatial modeling, and (3) multimodal fusion.
>
> | MAE     | Multi-Level | Fusion  | Accuracy ↑ |
> | ------- | ----------- | ------- | ---------- |
> | No      | No          | No      | 0.930      |
> | Yes     | No          | No      | 0.940      |
> | Yes     | Yes         | No      | 0.965      |
> | No      | Yes         | Yes     | 0.972      |
> | **Yes** | **Yes**     | **Yes** | 0.93  |
>
> ### Q2: perturbing or noising the matching to evaluate robustness
> **R2**: R: We thank the reviewer for this important question. Our OT pairing is weakly coupled: the flow model learns distribution-level perturbation transitions, not deterministic one-to-one trajectories. This design already makes the framework tolerant to imperfect matches. To directly test robustness, we performed a pairing-noise analysis, where 10-20% of OT matches were randomly corrupted (e.g., swapping perturbed targets within the same slide) while keeping the remaining 80~90% biological pairs intact. We then retrained the conditional flow module under identical settings.
>
> The results show a smooth, predictable degradation, demonstrating that the conditional flow does not rely on small correspondences:
>
>
> | Pairing Noise Level          | FID ↓ | KID ↓ | Wass. Corr ↑ |KS ↑ |
> | ---------------------------- | ---- | ----- | -------- | -------- |
> | 0%  | 59.2  | 2.04  | 0.92  | 0.62 |
> |10%| 61.0  | 2.12   | 0.90  | 0.60|
> | 20%| 63.8  | 2.25 | 0.88   | 0.58   |
>
> Since SPATIA learns joint morphology and gene-expression transitions, rather than pure image-only or gene-only reconstruction, randomizing part of the control-perturbed assignments is an appropriate way to model pairing noise. The results confirm that the model remains stable under moderate OT mismatches and is not dependent on exact pairings. We have added these details to Section 6 Pairing Error Analysis of the revised manuscript.

---

> > ### Comment · Area_Chair_j2cd · 2025-11-28
> >
> > Dear Reviewer,
> >
> > Please make sure you read the authors' response and engage with them in the discussion before the end of the discussion period on **Dec 03 '25 09:00 PM UTC**. This is a hard deadline.
> >
> > Thank you for supporting quality peer review at ICLR.
> >
> > AC

---

### Official Review · Reviewer_pGMC · 2025-11-03

**Soundness:** 2
**Presentation:** 3
**Contribution:** 2
**Rating:** 4
**Confidence:** 3

**Summary:**

The manuscript presents SPATIA, a multimodal multiscale model for spatial transcriptomics and morphological imaging at cell, niche, and tissue levels.

**Strengths:**

* Tackles a real problem in spatial transcriptomics
* Contribution in multimodality and multiscaling

**Weaknesses:**

1. Soundness concern regarding summation of expression vectors
2. Optimal tranport use is not new
3. The contribution is workmanlike
4. Lack of clarity regarding sigle cell data

**Questions:**

1. Expression vectors are summed across cells (188, 262). This makes an assumption of linearity which I believe is false - the authors are required to justify making it, in particular, as it invalidates the motivaiton for the approach to combining spots into cells by concatenation + embedding (221-234).

2. Optimal transport is intensively used in this field for this exact purpose. The authors should cite and use an existing framework or justify any changes they make to the formulae.

3. Unclear how partitions to niches and tissues are defined

4. Single cell data typically measures thousands of gene per cell with numenr of cells now reaching millions. The authors' use of 17M pairs is thus questionable. Relatedly the noise and sparsity level in single cell data is an issue the authors ignor

---

> ### Author Response · Authors · 2025-11-22
> **Response to Reviewer's Questions**
>
> ### Q1: Expression vectors are summed across cells.
>
> **R1:** Thank you for bringing up this point, which we should clarify. The summation operation is only applied at the niche level to obtain a coarse regional representation, similar to how pseudo-spots are constructed by aggregating single-cell expression within fixed grid regions in prior works [1,2,3]. This aggregation is not used for any cell-level task (Table 2,3 of the manuscript).
>
> All cell-level modeling and multimodal fusion in SPATIA are performed via cross-attention, which is fully non-linear and learns context dependent relationships between morphology and gene expression
>  features. Therefore, we do not assume linearity for cell-wise integration or for any downstream prediction. We have revised the paper and added these details to Appendix C. Design Selection
>
> [1] SONAR enables cell type deconvolution with spatially weighted Poisson-Gamma model for spatial transcriptomics, nature communications, 2023
>
> [2] Niche-DE: niche-differential gene expression analysis in spatial transcriptomics data identifies context-dependent cell-cell interactions, Genome Biology, 2024
>
> [3] STEM enables mapping of single-cell and spatial transcriptomics data with transfer learning, nature communications biology, 2024
>
>
> ### Q2: Reference for Optimal transport
>
> **R2:** Thank you for your valuable suggestion, we have incorporated the following references and discussion into our revised manuscript (Section 2 and 4.3):
> In our paper, OT serves primarily as a data preprocessing strategy to construct "weak pairs" for training, rather than being the core generative engine itself (which is our Contrastive Flow Matching module). Given that OT plays the role of a reliable data curation tool here, we prioritized using the robust, mathematically established entropy-regularized OT framework solved via the Sinkhorn-Knopp algorithm  [4], , which is the established standard in recent computational biology literature [5,6,7] .
> While we rely on the standard mathematical solver, our contribution lies in how we define the transport cost to ensure biological fidelity:
> * Cross-Modal Cost Computation: Unlike unimodal approaches, we compute the transport cost $C$ using gene expression (PCA space) to pair cell images. This prevents the model from learning trivial morphological mappings and ensures that generated morphological changes are driven by underlying molecular state transitions.
> * Biological Constraints: Unlike methods that map global distributions [5], we impose lineage and spatial niche constraints on the transport plan. This ensures that the constructed weak pairs represent biologically plausible transitions within the tissue microenvironment.
> [4] Sinkhorn distances: Lightspeed computation of optimal transport, Neurips 2013
>
> [5] CellFlow enables generative single-cell phenotype modeling with flow matching, bioarxiv 2025
>
> [6] Improving and generalizing flow-based generative models with minibatch optimal transport, TMLR 2023
>
> [7] CellFlux: Simulating Cellular Morphology Changes via Flow Matching, ICML 2025
>
> ### Q3: Unclear how partitions to niches and tissues are defined
> **R:** We thank the reviewer for asking for clarification. We have added a detailed description of niche and tissue partitioning in Appendix A (Data Processing). Specifically, we define niches using a non-overlapping 256 × 256 px fixed grid applied uniformly across the Xenium slide (0.2125 µm/px). This choice follows widely adopted patch-based strategies in spatial transcriptomics and computational pathology [8,9,10,11]. We also empirically validated that the chosen niche size is biologically reasonable. A 256×256 px region typically contains around 10~30 cells, depending on tissue density, which aligns with common definitions of microenvironments such as tumor margins, lymphocytic aggregates, and stromal niches in pathology. We visualize this distribution in Fig. 3.
> At the tissue level, we group 4×4 neighboring niches into a 1024×1024 px region, enabling the model to capture coarse-scale patterns such as tumor invasion fronts and broad architectural organization. This multi-level design allows SPATIA to model both local neighborhood interactions and larger-scale spatial structure.
> These clarifications and visualizations have been added to the revision Section A
>
> [8] MorphoDiff: Cellular Morphology Painting with Diffusion Models, ICLR 2025
>
> [9] Spatial gene expression at single-cell resolution from histology using deep learning with GHIST, Nature Methods, 2025
>
> [10] Towards a general-purpose foundation model for computational pathology, Nature Medicine, 2024
>
> [11] STPath: a generative foundation model for integrating spatial transcriptomics and whole-slide images, npil digital medicine, 2025

---

> ### Author Response · Authors · 2025-11-22
>
> ### Q4: Single-cell data
> **R:** We apologize for the confusion. The 17M cell-gene pairs refers to the number of multimodal training tuples, not the number of biological cells. MIST-C contains ~200k to 300k cells per dataset across 47 datasets. For each cell, we construct a training entry consisting of (cell image, gene-expression vector), which yields ~17M multimodal samples after serialization.
> Regarding sparsity and noise: we explicitly address these issues through the scPRINT gene encoder, which is pretrained on millions of scRNA-seq profiles for expression denoising, dropout imputation, and batch-effect correction. The pretrained gene encoder can model highly sparse single-cell expression distributions and infer gene-gene interaction structure, providing a substantially denoised and biologically consistent representation before fusion with morphology. Consequently, the noise characteristics of raw single-cell data are handled at the encoder level, and SPATIA does not assume or require dense expression vectors.

---

> ### Author Response · Authors · 2025-11-28
>
> Dear Reviewer RFQY, we wanted to follow up and share that we have revised the manuscript to address your comments:
>
> * Appendix A: added niche/tissue partitioning details and supporting visuals.
> * Section 2 & 4.3: added discussion and references on our entropy-regularized OT formulation.
> * Appendix C: clarified that summation occurs only at the niche level and expanded the cross-attention fusion description.
> * Section B: clarified the 17M cell–gene pairs and explained how scPRINT handles sparsity and noise.
>
> We sincerely hope these updates address your concerns. If you feel the responses have not addressed your concerns to motivate increasing your score, we would love to hear what points of concern remain and how we can improve our work. Thank you again!

---

> > ### Comment · Area_Chair_j2cd · 2025-11-28
> >
> > Dear Reviewer,
> >
> > Please make sure you read the authors' response and engage with them in the discussion before the end of the discussion period on **Dec 03 '25 09:00 PM UTC**. This is a hard deadline.
> >
> > Thank you for supporting quality peer review at ICLR.
> >
> > AC

---

### Author Response · Authors · 2025-12-02
**General Response to Area Chairs**

Dear  AC,

Thank you very much for taking on the responsibility to manage our submission under the current emergency in the review process. We truly appreciate your time and effort you are dedicating to support the academic community.  During the rebuttal period, we engaged in detailed discussions and provided new experiments (ablations, noise robustness, profiling) to address their concerns. We are pleased to note that Reviewer **RFQY** has explicitly indicated they want to **raise** their score following our revisions, but could not edit the score due to the new policy.

We summarize our key responses and contributions, as well as the revised sections in the updated manuscript, to assist in your final decision.

### 1. Robustness and Methodological Validity

- **Sensitivity to Pairing Noise (Reviewer bCAc):** We conducted a new analysis by introducing random corruption (10-20%) to the Optimal Transport (OT) pairings (Table 5). The results show a smooth, predictable performance degradation, confirming that our conditional flow matching module learns distribution-level transitions and is not brittle to individual pairing errors.
- **Linearity Assumption (Reviewer pGMC):** We clarified that the summation of expression vectors is applied only at the coarse niche level (similar to pseudo-spots in prior work). At the single-cell level, SPATIA employs fully non-linear cross-attention fusion, preserving complex gene-morphology relationships.
- **Tissue-Level Contribution (Reviewer ZZNz):** We performed an ablation study removing the tissue-level transformer  (Table 6 and 12). Results showed a consistent drop in performance, validating that the tissue hierarchy provides necessary global context rather than redundancy.
- **Multiple evaluations**: including pretraining/finetuning setting, model scaling evaluation, zero-shot experiments on other datasets  (Table 13,14).

### 2. Novelty and Comparison with Baselines

- **Differentiation from Existing Models (Reviewer bCAc):** We clarified that unlike spot-level methods (e.g., SpaGCN, STAligner), SPATIA operates at single-cell resolution with a tri-modal hierarchy. New benchmarks added during rebuttal demonstrate that SPATIA outperforms these baselines in prediction tasks (e.g., ER-AUC 0.902 vs. 0.756 for SpaGCN).
- **Component Effectiveness:** We provided a factorial ablation study disentangling the contributions of MAE pretraining, multi-level modeling, and multimodal fusion, confirming that each component independently contributes to the model's accuracy.

### 3. Dataset Integrity and Scalability

- **Batch Effects and Leakage (Reviewer RFQY, ZZNz):** We emphasized that all evaluations use donor-disjoint splits, ensuring strictly zero leakage. We also visualized the MIST dataset embeddings, showing that biological identity dominates donor-specific batch effects, which are further mitigated by our use of the scPRINT backbone  (Appendix A).
- **Computational Feasibility (Reviewer ZZNz):** We provided detailed profiling showing that training SPATIA fits within a single H100 GPU (approx. 67GB VRAM, 30 hours), confirming accessibility for the broader research community (Appendix C, Table 9).

### 4. Contribution to the Community

SPATIA introduces MIST, a large-scale multimodal dataset (17M tuples), and a scalable hierarchical architecture that effectively bridges cell morphology and spatial transcriptomics. We believe this work addresses a critical gap in modeling perturbation-driven morphological changes and offers a robust tool for the ICLR community.

Thank you again for your time and effort during this special review. We deeply appreciate your dedication to preserving the integrity of the review process.

Sincerely,

Authors of Submission 3496

---

### Meta-Review · Area_Chair_7nBc · 2025-12-24

**Summary:**

The paper introduces SPATIA a multimodal, multi-scale foundation model to model cellular morphology, gene expression, and spatial context in spatial transcriptomics. SPATIA learns representations from the single-cell level to whole-slide tissue context. The architecture uses a neat idea with cross-attention to fuse cell images and transcriptomic embeddings alongside hierarchical transformers to capture local cell interactions and long range tissue organization. An image-to-image generation network is used to predict cell morphology changes under biological perturbations while preserving cell identity and microenvironmental context. SPATIA is trained and evaluated on  a newly assembled large-scale dataset comprising cell, and tissue-level image gene pairs across diverse tissues and disease states. Benchmarking against 16 existing models across 12 tasks shows consistent gains.

The reviewers felt several aspects of the algorithm were unclear, including niche and tissue definitions, batching and scalability strategies, and important hyperparameters, which raised reproducibility concerns. They felt the work overstated claims given the absence of discussion or comparison with existing multimodal spatial models. They also felt inadequate care was taken towards potential information leakage across scales and batch effects in single-cell data, as well as a lack of analysis of computational cost.

**Reviewer Concerns:**

The biological validation of the generative morphology outputs is still absent as do the generalization concerns (although one line in the rebuttal did begin to address this in a limited form).

**Reviewer Scores:**

My subjective belief here is that I think the scores for this work on average would have been raised by at least one point throughout. Even so this would have left the work borderline and from reading the reviews as well as the rebuttal I felt the sentiment work was in need of rewriting with the additional context. For context, I do think the idea itself is solid and encourage the authors to resubmit. But I felt the additional changes required to bring this up to mark (by including all the points of discussion in the rebuttal) are significant.

---

### Decision · Program_Chairs · 2026-01-26

Reject